# Aborted propagation of the Ethiopian rift caused by linkage with the Kenyan rift

Giacomo Corti [1], Raffaello Cioni[2], Zara Franceschini[2], Federico Sani[2], Stéphane Scaillet[3], Paola Molin[4], Ilaria Isola[5], Francesco Mazzarini[5], Sascha Brune [6,7], Derek Keir[2,8], Asfaw Erbello[9], Ameha Muluneh[10], Finnigan Illsley-Kemp [8,11] & Anne Glerum[6]

Continental rift systems form by propagation of isolated rift segments that interact, and eventually evolve into continuous zones of deformation. This process impacts many aspects of rifting including rift morphology at breakup, and eventual ocean-ridge segmentation. Yet, rift segment growth and interaction remain enigmatic. Here we present geological data from the poorly documented Ririba rift (South Ethiopia) that reveals how two major sectors of the East African rift, the Kenyan and Ethiopian rifts, interact. We show that the Ririba rift formed from the southward propagation of the Ethiopian rift during the Pliocene but this propagation was short-lived and aborted close to the Pliocene-Pleistocene boundary. Seismicity data support the abandonment of laterally offset, overlapping tips of the Ethiopian and Kenyan rifts. Integration with new numerical models indicates that rift abandonment resulted from progressive focusing of the tectonic and magmatic activity into an oblique, throughgoing rift zone of near pure extension directly connecting the rift sectors.

[1] Istituto di Geoscienze e Georisorse, Consiglio Nazionale delle Ricerche, Florence 50121, Italy. [2] Dipartimento di Scienze della Terra, Università degli Studi di Firenze, Florence 50121, Italy. [3] Institut des Sciences de la Terre d'Orléans (ISTO), Université d'Orléans CNRS BRGM, Orléans 45071, France. [4] Dipartimento di Scienze, Università degli Studi di Roma Tre, Rome 00146, Italy. [5] Istituto Nazionale di Geofisica e Vulcanologia, Sezione di Pisa, Pisa 56126, Italy. [6] German Research Centre for Geosciences GFZ, Potsdam 14473, Germany. [7] Institute of Earth and Environmental Science, University of Potsdam, Potsdam 14476, Germany. [8] School of Ocean and Earth Science, University of Southampton, Southampton SO14 3ZH, UK. [9] School of Applied Natural Sciences, Adama Science and Technology University, Adama 1888, Ethiopia. [10] School of Earth Sciences, Addis Ababa University, Addis Ababa 1176, Ethiopia. [11] School of Geography, Environment and Earth Sciences, Victoria University of Wellington, Wellington 6012, New Zealand. Correspondence and requests for materials should be addressed to G.C. (email: giacomo.corti@igg.cnr.it)

Continental rift systems do not form single, straight structural entities but comprise a series of discrete major segments that kinematically link during progressive extension[1–3]. The regions of rift interaction usually coincide with offsets, kinks or major changes in the trend of adjacent rift segments[1], which may reflect a control exerted by the pre-existing structural grain on rifting[4]. The growth of major segments of continental rifts and their mechanical linkage influence many aspects of rifting, including volcanism, seismicity, topography and depositional environments[5]. Interaction between rift segments has also been suggested to control the segmentation of mid-ocean ridges, with strike-slip transfer faults connecting offset rifts representing possible precursors of oceanic transform faults[6–8]. Despite its importance, the complex, three-dimensional evolution of rift linkage is poorly documented in active continental rifts and difficult to accurately reconstruct from rifted conjugate margins. Here we shed light on these processes by providing new detailed constraints on the volcanic and tectonic activity of the Ririba rift (southern Ethiopia) and elucidate the complex spatial and temporal evolution of interaction between the Ethiopian and Kenyan rifts in the Turkana depression.

The Ririba rift[9–11] is the southern termination of the Ethiopian rift, a segment of the East African rift which results from the ongoing separation between the African and Somalian plates, at rates of 3–5 mm per year in a roughly E-W direction[12] (Fig. 1). This rift is part of the Turkana depression, a lowland located between the uplifted East African and Ethiopian plateaus characterised by a complex deformation pattern, which results from the interaction between the Ethiopian and Kenyan rifts (Fig. 1). In striking contrast to the narrow (<100 km) rift valleys of Ethiopia and Kenya, the Turkana depression is marked by a wide region of ongoing rift-related tectono-magmatic activity, where faulting, seismicity and Quaternary-Holocene volcanism are spread over a width of more than 300 km (ref. [13]; see Supplementary Note 1). Within this wide region of tectonic activity, extension is clustered within different sub-parallel regions of deformation, corresponding to the basin hosting Lake Turkana, the Kino Sogo fault belt and the Ririba rift itself (Fig. 1). The Lake Turkana basin corresponds to the northwestward propagation of the Kenyan rift via the Suguta valley, whereas the Ririba rift is suggested to result from the southward propagation of the Ethiopian rift[13–15]. Between the two regions, the Kino Sogo fault belt is a 30-km-wide system of small horsts and grabens, believed to have accommodated a minor part of regional extension[15,16]. To the north, this fault system links with the Chew Bahir basin and the Gofa Province, which are part of the complex broadly rifted zone of South Ethiopia[10,13].

Numerical and physical modelling of rift linkage suggests that the anomalous breadth of the volcanic and tectonic activity in the Turkana depression has been likely controlled by the presence of a pre-existing heterogeneity in the Turkana lowland, corresponding to a system of NW-SE-trending Mesozoic-Paleogene grabens[17] transversal to the trend of the Ethiopian and Kenyan rifts (Supplementary Note 1). Complex patterns of migration of deformation are believed to contribute to wide deformation in the Turkana depression, where faulting has been suggested to have propagated from West (e.g., Lokitipi, Lokichar basins) to East (e.g., Lake Turkana basin, Kino Sogo belt) during the Oligocene-Recent, reaching the Ririba rift in the Quaternary[13] (Supplementary Note 1; Supplementary Figs. 1, 2). Therefore, this rift was previously believed to accommodate current strain related to both West to East, as well as North to South, propagation of deformation[13,14]. However, no detailed information on the timing of fault activity in the Ririba rift has previously been available. To address this, we conducted geological–structural and analytical (geochronological and geochemical) work on samples collected in the Ririba rift to characterise its tectonic and volcanic activity. Our data place important constraints on how the Ethiopian and Kenyan rifts have interacted and are now linking. The results suggest that the southward propagation of the Ethiopian Rift to form the Ririba rift during the Pliocene was short-lived and aborted close to the Pliocene-Pleistocene boundary, with the Ethiopian and Kenyan rifts instead directly linking by a narrow, throughgoing rift zone in the Turkana depression.

## Results

**Characteristics of the tectonic activity in the Ririba rift**. The extensive remote sensing analysis and the fieldwork in the area indicate that extensional deformation is accommodated by a complex system of normal faults, which define the narrow (10–25 km wide), roughly N-S rift (Fig. 2). This rift is bounded by a few major normal faults located on the western side, giving rise to a mostly asymmetric architecture of deformation (Figs. 2b and 3; Supplementary Fig. 3). These structures have a dominant dip-slip displacement, consistent with a local extension direction of ~N95°E (Fig. 3), in line with the regional plate kinematics[12]. The NNW-SSE-trending boundary faults are long (upto >40 km), rather linear and, despite their length, are characterised by limited vertical offset (typically <50 m; Fig. 2; Supplementary Fig. 4; Note 2). Analysis of the cumulative vertical offset of faults indicates maximum values of ~160 m, which corresponds to a horizontal displacement of ~60–95 m considering average fault dips of 60–70° (ref. [16]). In turn, this indicates that the faults have accommodated a limited amount of extension (maximum stretching factors $\beta$ of ~1.004). Analysis of displacement on major boundary faults indicates a decrease of vertical offset of one order of magnitude passing from the Segen basin (>500 m) to the Ririba rift (Fig. 2b). This is accompanied by a decrease in basin width from ~30 km in the Segen basin to ~10 km in the Ririba rift. This progressive southward decrease in boundary fault displacement and rift width is consistent with the Ririba rift representing the southern propagation of the Ethiopian rift in the Turkana depression via the Segen basin[13,14].

The overall fault pattern (excluding the major faults) is characterised by interaction between variably oriented normal faults. This is exemplified at the southern termination of the rift where NNW-SSE and roughly NE-SW structures intersect at a high angle (Fig. 2). Similar complex interactions among different sets of faults (with NNE-SSW, NE-SW, or ENE-WSW orientations) are also observed in the area to the East of the Ririba rift, at the northeastern termination of the Hurri Hills (Fig. 2c). This angular pattern, the anomalously low displacement/length ratio, the shape of the displacement/length curve of major faults and the trend of rift-related structures not orthogonal to the extension direction but parallel to basement fabrics (Supplementary Note 2; Supplementary Fig. 5) are suggestive of a strong influence of the pre-existing structures on fault development, as observed in the nearby Kino Sogo fault belt[15,16] and Chew Bahir basin[18,19].

**Volcanic activity**. Tectonic activity in the Ririba rift has been accompanied by widespread volcanic activity of mostly basaltic (sensu lato) composition[20–24]. Apart from Middle Miocene alkali basalts and hawaiites with K/Ar ages in the range of 12.3–10.5 Ma (ref. [23]) occurring north of the Ririba rift, volcanism was characterised by two phases of Late Miocene-Pliocene and Quaternary activity. The first phase corresponds to voluminous effusion of basaltic andesites or tholeiites (Bulal basalts of refs. [21,22]) forming a basal widespread, relatively thin (thickness of a few tens of metres[15,22]) lava platform, with reported K/Ar ages ranging from

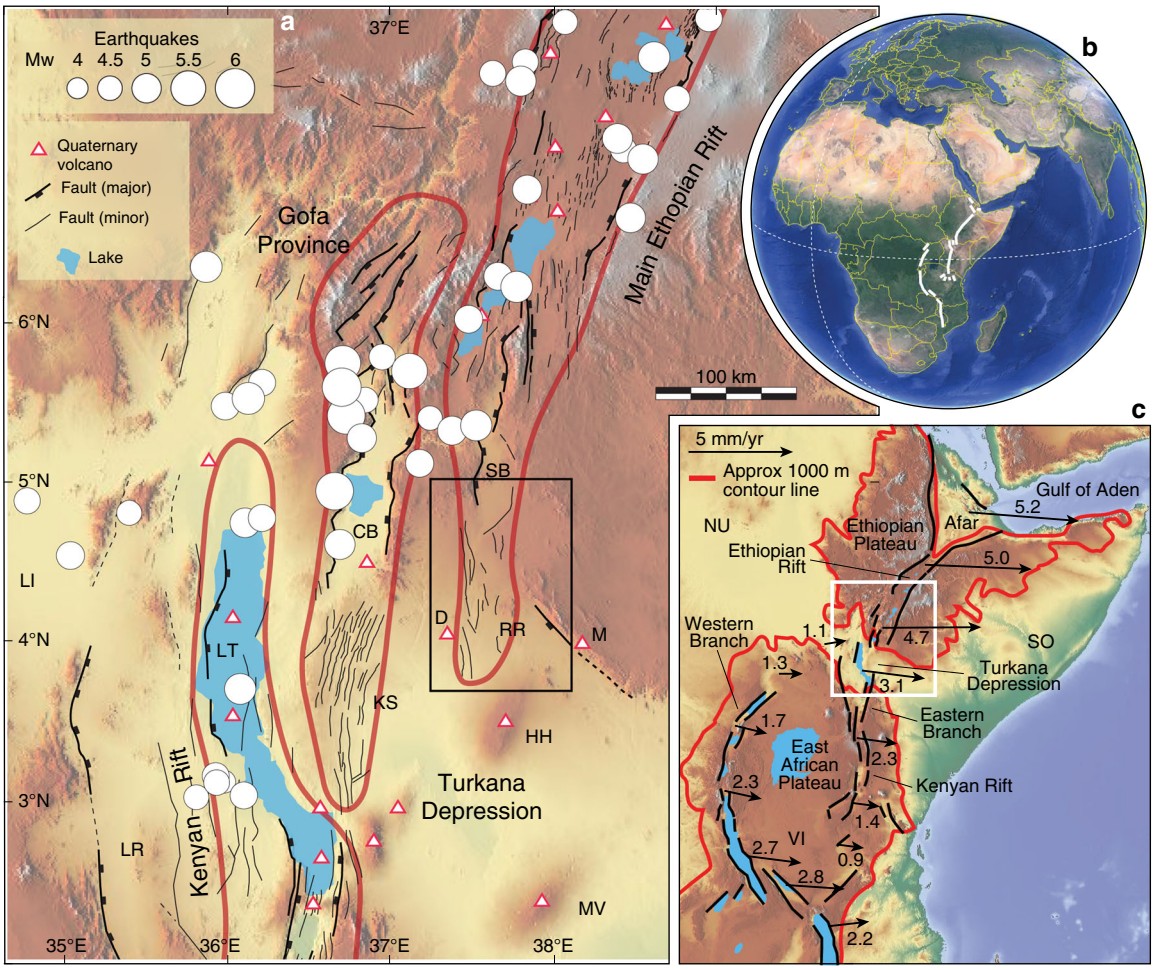

**Fig. 1** Tectonic setting of the Turkana depression. **a** Quaternary faults, seismicity and Quaternary volcanoes in the Turkana depression and surrounding regions superimposed on a SRTM (NASA Shuttle Radar Topography Mission) digital elevation model (modified from ref. [17]). Seismicity from USGS National Earthquake Information Center (NEIC) catalogue (https://earthquake.usgs.gov/earthquakes/search/); Quaternary volcanoes from the Smithsonian Institution, Global Volcanism Program[62] (http://volcano.si.edu/). Red lines delineate the deformation domains of the Kenyan rift, Kino Sogo fault belt (KS)-Gofa Province, and Main Ethiopian rift[13, 15]. Black square indicates the area illustrated in Fig. 2. CB: Chew Bahir basin; D: Dilo-Dukana volcanic field; HH: Hurri Hills; LI: Lokitipi basin; LT: Lake Turkana; LR: Lokichar basin; M: Mega volcanic field; MV: Marsabit volcano; SB: Segen basin; RR: Ririba rift. **b** Location of the East African rift (white lines); **c** schematic fault pattern and present-day plate kinematics. Black arrows show relative motions with respect to a stable Nubian reference frame according to the best-fit model of ref. [12]. Values besides arrows indicate motion in mm per year. White square indicates the area portrayed in panel **a**. NU: Nubian plate; SO: Somalian plate; VI: Victoria microplate

6.1 to 3.6 Ma (refs. [21,23]). New Ar/Ar dating of two samples (MDZ05, MDZ12) indicates ages of 3.69–3.75 Ma for these tholeiitic basalt lavas (Fig. 4; Supplementary Figs. 6–9; Supplementary Note 3; Supplementary Tables 1,2), consistent with the age and composition of the Bulal basalts in Kenya.

The second phase is characterised by numerous small-volume eruptions of alkaline, generally mafic lavas forming the alignments of cinder/scoria cones, maar craters, and alkaline basaltic lava flows often entraining abundant mantle and crustal xenoliths of the Dilo-Dukana volcanic field. Ages for this activity range between 0.9 and 0.3 Ma (ref. [23]). New Ar/Ar dating on three nepheline basanite lava flows in the Ririba rift (MDZ03, MDZ10, and MDZ13) indicates tightly clustered ages of 0.13–0.16 Ma (Fig. 4; Supplementary Fig. 6; Supplementary Table 1). The age and composition (Supplementary Figs. 8, 9; Supplementary Note 3; Supplementary Table 2) of these lavas is consistent with the Quaternary activity of the Dilo-Dukana, Hurri Hills, and Mega volcanic fields (Fig. 2).

Statistical analysis of the distribution of the Quaternary volcanic vents indicates an overall, NE-SW elongation of the

Dilo-Dukana volcanic field, with a complex distribution of the vents at a more local scale. The overall elongation of the field, highly oblique to the rift trend but subparallel to a major pre-existing structure (the Buluk fault zone) west of the Ririba rift[15,16] (Supplementary Note 4), is suggestive of a strong control exerted by inherited basement fabrics on the distribution of volcanic vents. This is supported by distribution of vents at local scale, which is controlled by the angular network of NNE-SSW, NE-SW, or ENE-WSW structures that reactivate pre-existing fabrics (Supplementary Fig. 10; Supplementary Note 4).

**Timing of rifting.** The main boundary faults of the Ririba rift offset the ca. 3.7 Ma lavas, which correlates to the Bulal basalts of refs. [21,22] (Fig. 2). The morphology of this widespread, relatively thin lava platform is best explained by it having erupted along fissures onto a relatively flat surface without significant surface expression of faulting. Therefore, we consider the rift to have potentially developed concurrently with, but most likely after, this Pliocene volcanic event. Consequently, we consider the main

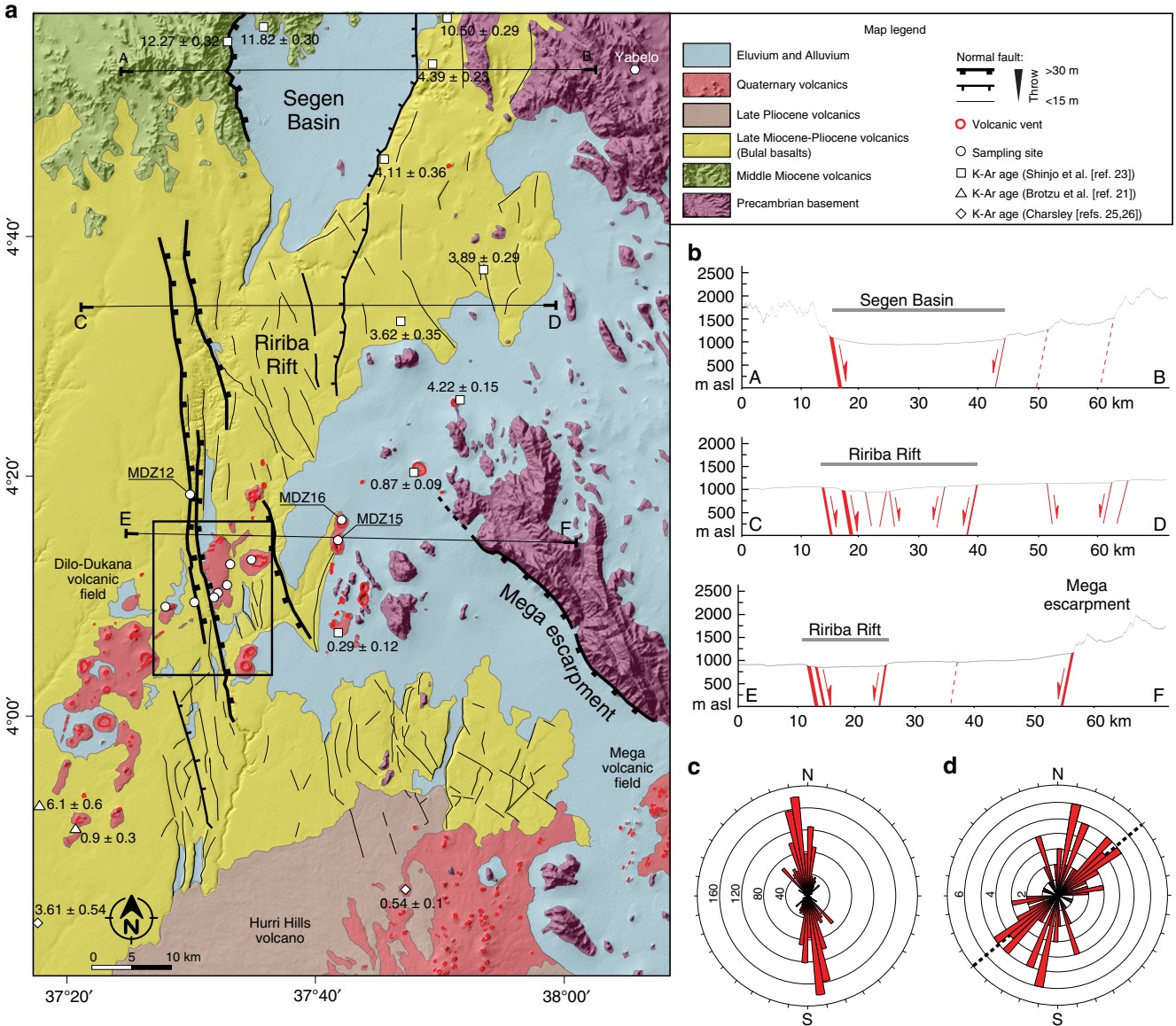

**Fig. 2** Geology of the Ririba rift. **a** Geological map superimposed on a SRTM (Nasa Shuttle Radar Topography Mission) digital elevation model. White dots labelled 'MDZ + number' indicate rock samples. Black square indicates the area illustrated in Fig. 3. **b** Simplified structural cross-sections across the Segen Basin (A-B) and the Ririba rift (C-D, E-F). **c** Rose diagram illustrating plots of weighted fault azimuths. The weighting factor for each fault is the ratio between the length and the minimum length of the whole data set, such that long faults have higher ratio (weight) than short ones. The frequency of the azimuth of a fault directly relates to this ratio, the longer the fault the higher its frequency. **d** Rose diagram illustrating plots of the elongation of volcanic vents; the dashed black line indicates the overall elongation of the Dilo-Dukana volcanic field

boundary faults to be representative of the extension-related deformation for the entire history of the Ririba rift.

These boundary faults are buried by two 0.13–0.16 Ma lava flows in two different places (labelled as A and B in Fig. 3), where large flows, which partly filled the rift depression, forced the Ririba River to shift to the West and to incise a narrow valley partially eroding the fault footwall (see also Supplementary Fig. 11). The overall NE-SW trending alignment of the volcanic vents indicates that the feeding structures of these Quaternary volcanoes obliquely cut the N-S Ririba faults, whose Pleistocene inactivity is further supported by several other lines of geological and morphological evidence (Supplementary Note 5). Importantly, normal faults at the southern termination of the rift crosscut the ~3.7 Ma Bulal lavas but are sealed by the basal flows of the large shield volcano of the Hurri Hills (Fig. 2; Supplementary Fig. 12; Supplementary Note 5). These flows,

with a poorly constrained oldest age of 2.8 Ma (refs. [25,26]), are weakly affected by few minor normal faults at the northern termination of the shield, while they are overlain by undeformed younger products with ages as old as 2.3 Ma (ref. [21]; Supplementary Fig. 12; Supplementary Note 5). This constrains the end of rift activity to have likely occurred close to the Pliocene-Pleistocene boundary.

## Discussion
In line with previous findings[13,14], these results indicate that the Ethiopian rift propagated southwards, into the Turkana depression, to form the Ririba rift during an extensional phase likely coinciding with, and certainly following, the emplacement of the large lava platform at ~3.7 Ma. However, in striking contrast with previous suggestions[13,14], this propagation aborted and the rift

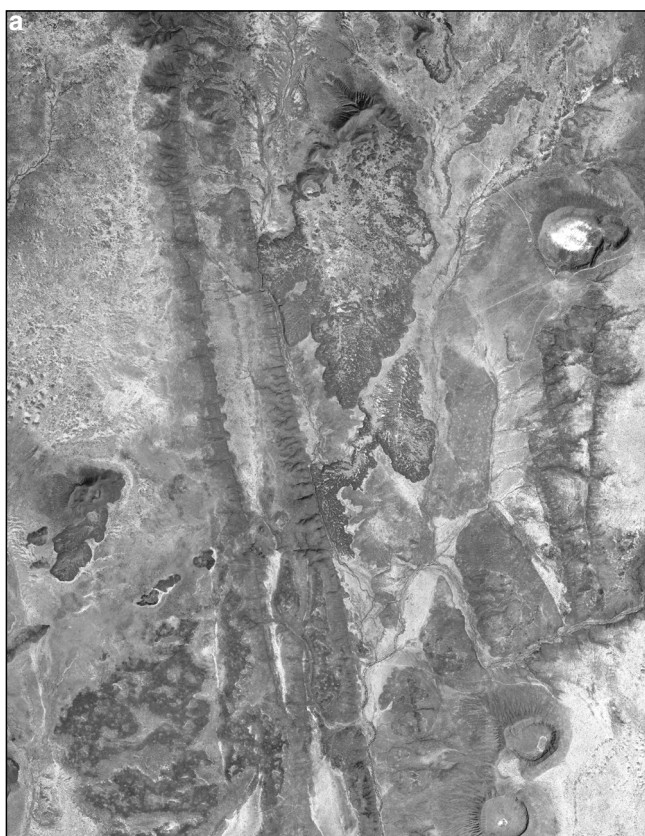

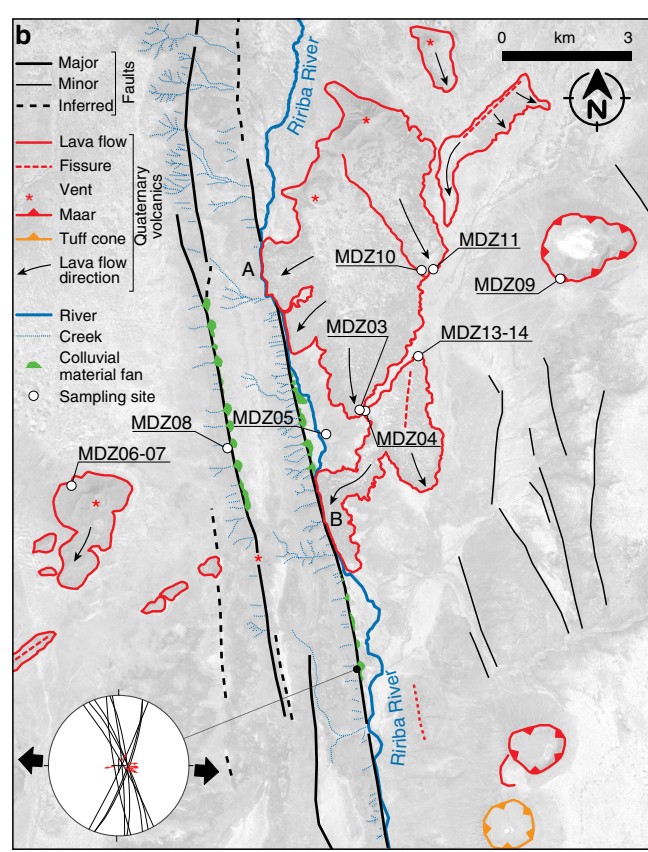

**Fig. 3** Quaternary volcanic features, faults and morphotectonic elements of the central part of the Ririba rift. Satellite image (courtesy of the DigitalGlobe Foundation) (**a**) and line drawing of Quaternary volcanic features, faults and morphotectonic elements (**b**). Letters A and B denote locations discussed in the text. Plot in the bottom left corner shows stereographic projections (Wulff net, lower hemisphere) of the fault planes and the collected associated fault-slip vectors (from the station of structural measurements indicated with the black dot); black arrows indicate the direction of extension obtained by using the P&T axes inversion method calculated with Win-Tensor[63]

was already abandoned at the beginning of the Pleistocene (Fig. 5). Therefore, the Ririba rift accommodated a limited amount of extension-related deformation in a rather short time interval (possibly <1.5 Ma), in a similar fashion to what could have happened in the nearby Kino Sogo fault belt[16]. A relatively short history of localised rifting in the Ririba rift is consistent with very little crustal thinning and thin accumulation of sediments/volcanic products in the area interpreted from regional wide-angle seismic, gravity and receiver function data, and global sediment thickness data[27].

Morphotectonic analysis of the Mega escarpment (Fig. 2; Supplementary Fig. 13; Supplementary Note 6) indicates absence of recent-current tectonic activity, which argues against an eastward propagation of deformation. Instead, distribution of Holocene volcanism, earthquakes recorded on global seismic networks and analysis of seismic moment release (Fig. 5) indicates that the volcanic and tectonic activity is currently focused to the West, in the Lake Turkana basin, and in the region North of the Ririba rift (Gofa Province), forming a continuous narrow belt of extension connecting the Kenyan and Ethiopian rifts[13]. This interpretation is consistent with the locus of strain and of the plate boundary deduced from past temporary, local seismic networks deployed in northern Kenya[28], and from the modelling of geodetic data in East Africa[9,29,30] (Fig. 5). This suggests that abandonment of the Ririba rift resulted from the progressive narrowing of the deformation in the Turkana depression and the direct linkage between the two offset major rift sectors in South Ethiopia (Fig. 5).

To corroborate this evolution, we have run new numerical models that reproduce the 3D rift evolution of interaction

between the offset Kenyan and Ethiopian rifts at a much higher resolution with respect to previous simulations of the region[17]. The models include two offset narrow rift valleys (Ethiopian and Kenyan rifts) separated by a transversal pre-existing heterogeneity corresponding to the Turkana depression (Fig. 6a). In accordance with the configuration in nature, the Kenyan rift is characterised by a N-S trend, the Ethiopian rift strikes roughly NE-SW, and the transversal domain is modelled with a NW-SE orientation. Initial thicknesses of crust and mantle lithosphere need to reproduce mid-Miocene conditions before the onset of the current rift phase and are therefore based on geophysical data[27,31,32] in domains adjacent to Cenozoic rift activity (Fig. 6a). Due to radiogenic crustal heat production and thermal equilibration, the initially 30-km-thick crust in the Turkana depression results in a stronger lithosphere than in the Kenyan and Ethiopian rift areas with initially 40-km-thick crust (Fig. 6b). We account for tectonic inheritance that guided the Kenyan and Ethiopian rifts along crustal foliations and mantle weaknesses of pre-Miocene tectonic activity in a simplified manner: we implement a slightly thinner lithosphere beneath these rifts with the advantage of inducing strain localisation without imposing crustal fault locations. In contrast to a previous study[17], the presented model accounts for oblique rifting in the Main Ethiopian rift, which affects stress focusing in the study region. It also features a larger model domain of 1200×600×165 km and a higher resolution of up to 2.5 km, which allows for more detailed analysis of local extension direction during rift localisation in previous models[17]. The governing physical equations are solved using the geodynamic modelling software ASPECT[33-37]. A detailed description of

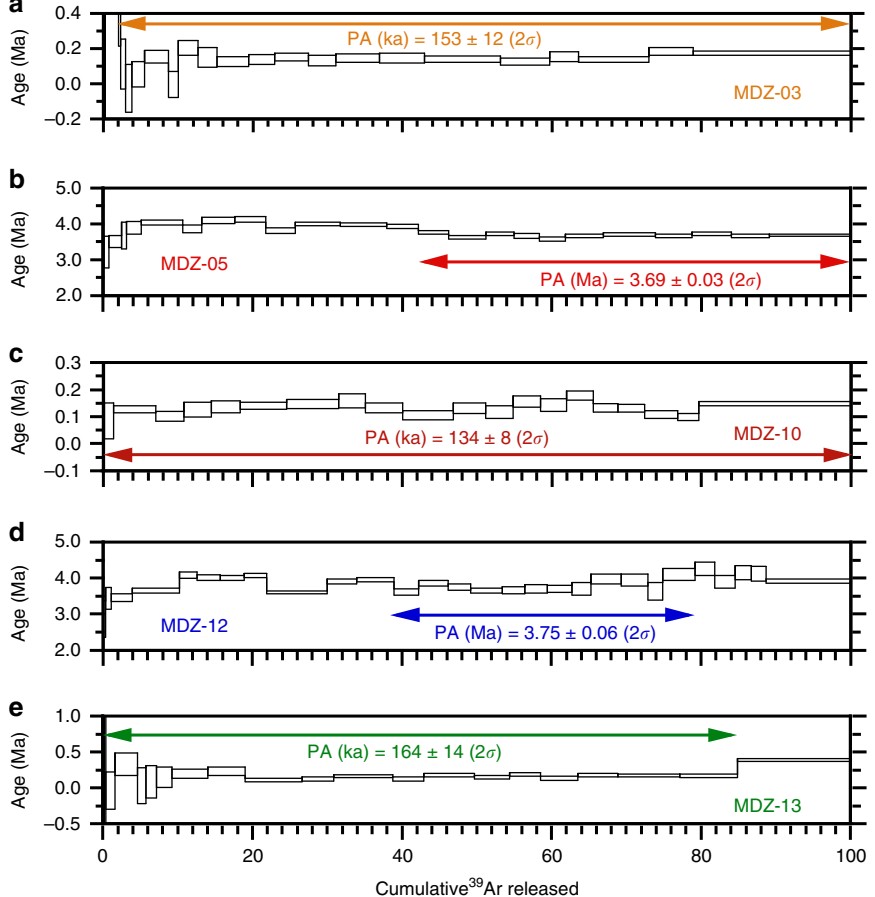

**Fig. 4** Ar/Ar analysis. Graphs illustrating the Ar/Ar data of samples MDZ03 (**a**), MDZ05 (**b**), MDZ10 (**c**), MDZ12 (**d**), and MDZ13 (**e**), reporting Plateau Age (PA) with the respective estimated error. Arrows span consecutive steps included in the plateau ages. Boxes represent the heating-steps, with the width of the box representing the percentage of $^{39}$Ar released and its thickness corresponding to the analytical error (lower and upper limits of the uncertainty at ±1 σ)

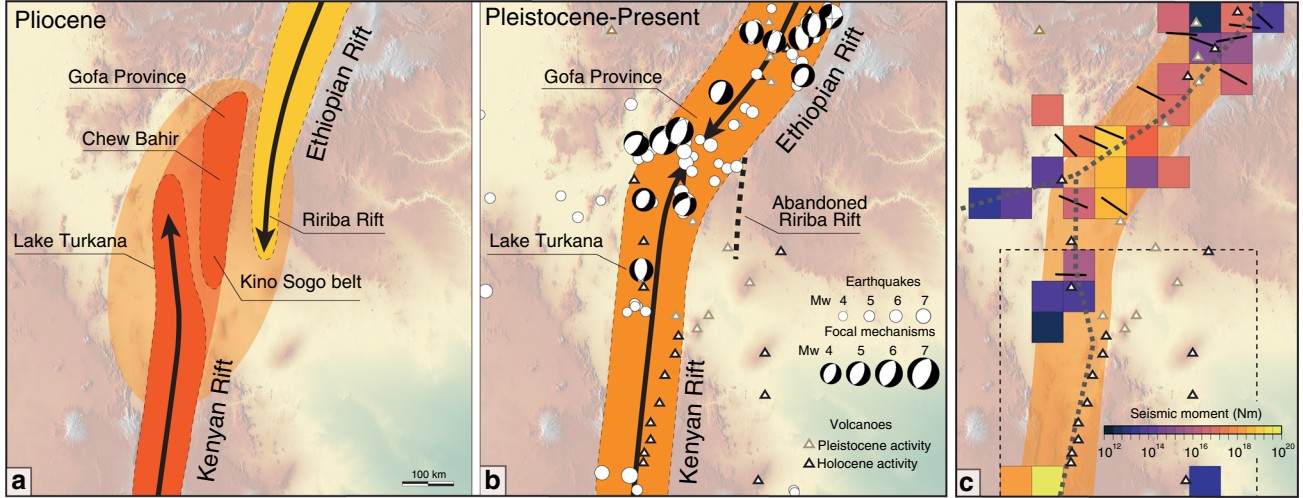

**Fig. 5** Evolution of rifting and current distribution of deformation in the Turkana depression. **a**, **b** Pliocene-recent schematic evolution of rifting. Coloured patterns in panel **a** delineate the main domains of tectonic and magmatic activity related to the Ethiopian and Kenyan rifts; the elliptical light orange pattern indicates the hypothesised extension of the region affected by deformation; black lines with arrows indicate the patterns of rift propagation[15]. Earthquakes are from the NEIC catalogue and focal mechanisms from the Global Centroid-Moment-Tensor (GCMT) Project[64,65] (https://www.globalcmt.org). Dotted black line indicates the abandoned Ririba rift. **c** Sum of seismic moment release of earthquakes from the NEIC catalogue; superimposed are T-axes strike directions from the GCMT moment tensors (black lines). Dashed box indicates the region analysed by the temporary, local seismic networks deployed in northern Kenya[28]. Dotted dark grey line indicates the approximate location of the plate boundary deduced from models of geodetic data in East Africa[12,29,30]

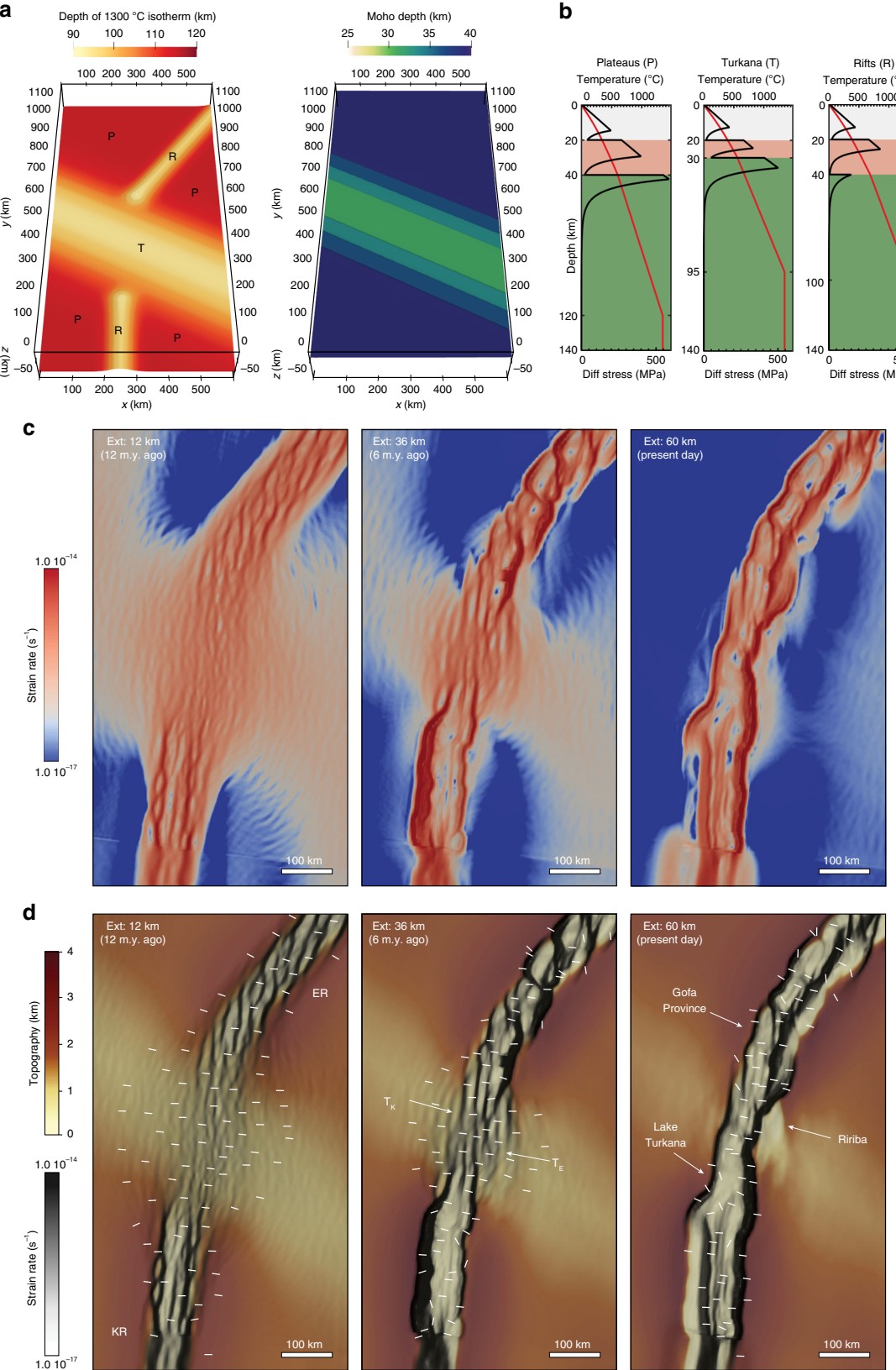

thermal and rheological configurations, mesh refinement strategy as well as initial and boundary conditions can be found in the Methods section.

Our model results illustrate a progressive narrowing of the region undergoing deformation for increasing extension (Fig. 6c),

from a wide region of diffuse faulting and thinning (width >300 km) to a throughgoing narrow rift valley (width <100 km) directly linking the two major rift segments (see also Supplementary Fig. 14; Supplementary Note 7). The initial rift width in the area of the Turkana depression is not related to crust-mantle

**Fig. 6** Numerical modelling of rift interaction in the Turkana depression. **a** Initial depth distribution of the 1300 °C isotherm and the Moho. Three distinct model domains are labelled as follows: P: Plateaus; R: Ethiopian and Kenyan rifts; T: Turkana depression. **b** Geotherms (top axis, red lines) and yield strength profiles (bottom axis, black lines) in the different model domains. Yield strengths have been computed for the initial bulk extension rate of the models, i.e. 2.7×10$^{-16}$ 1/s, in the different domains. **c** Evolution of the deformation pattern illustrated as second invariant of strain rate at different amounts of bulk extension. Note the progressive narrowing of deformation in the Turkana depression for increasing extension; in the latest stages of rifting, extension is strongly localised within a continuous narrow region of strain connecting the offset major rift segments. **d** Evolution of model surface topography; white bars show the direction of the minimum horizontal stress (local direction of extension). Semi-transparent strain rate field is shown for reference. Note the dominant WNW-ESE orientation of the minimum horizontal stress in the oblique (roughly NE-SW trending) transfer zone, which deviates from the imposed roughly E-W plate motion. ER: Ethiopian rift; KR: Kenyan rift; T$_E$ and T$_K$ indicate the overlapping tips of the Ethiopian and Kenyan rift valleys, respectively; also shown are the regions approximately corresponding to the Gofa Province, Lake Turkana basin and Ririba rift

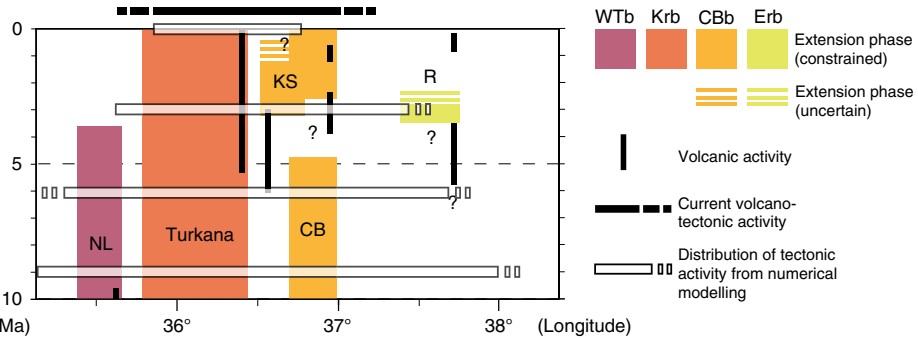

**Fig. 7** Spatio-temporal evolution of recent deformation in the Turkana depression and comparison with model predictions. Chronograms illustrating the spatial and temporal relationship of late Miocene-recent extension and volcanic activity. Reported are the current distribution of volcano-tectonic distribution from available geological and geophysical data, and the spatial distribution of tectonic activity at different time intervals as predicted from numerical modelling. Colours indicate the different deformation domains (WTb: West Turkana basins; Krb: Kenyan rift basins; CBb: Chew Bahir basins; Erb: Ethiopian rift basins). CB: Chew Bahir; KS: Kino Sogo; NL: North Lokichar; R: Ririba

decoupling[38], but to rift linkage dynamics in a domain without favourable oriented inherited faults or foliations[17]. Successive localisation towards the present-day Lake Turkana rift reduces the width of the deforming area and ultimately results in the abandonment of the Ririba rift. This appears as a westward migration of the deformation towards the Kenyan rift in the Lake Turkana basin, which is directly connected to the Ethiopian rift via an oblique narrow transfer zone corresponding to the Gofa Province (Fig. 6c; Supplementary Fig. 14; Supplementary Note 7). The evolution of rifting in the models matches the spatio-temporal history of the volcano-tectonic activity in nature very well (Fig. 7). Interestingly, the deforming area during the early stage of the rift model is wider than inferred from field observations in the eastern part of the rift (Fig. 7). This suggests that diffuse deformation may have affected the future Ririba rift before the onset of localisation. Future surveys in this region are required in order to test this model prediction.

Our model shows that the local extension direction in the centre of this oblique transfer zone rotates from the regional roughly E-W trend imposed by plate kinematics to roughly WNW-ESE (i.e., approximately orthogonal to the rift trend in the transfer zone; Fig. 6d). This explains the almost pure dip-slip earthquake focal mechanisms in the Gofa Province, which indicate extensional deformation in a roughly WNW-ESE to NW-SE direction, implying a rotation of the stress field from the regional E-W Nubia-Somalia motion observed in the Ethiopian rift[39] and in the Lake Turkana basin[30] (Fig. 5). This is also consistent with the local variation in the fault pattern and kinematics observed in the Gofa Province[19].

One interesting feature of this evolution is that the latest (Quaternary) phase of volcanism related to the Dilo-Dukana field is temporally and structurally unrelated to the main rifting event. This volcanic phase occurred after rift-related deformation ceased, with the composition (more alkalic character) and

volcanological features (important decrease in the volume of erupted products) well correlating with the decrease in tectonic activity in the Ririba area. The occurrence of abundant mantle xenoliths in all the products of the Quaternary volcanic vents outcropping in the area[24] suggests a rapid ascent of magma along deep feeding structures, which likely exploit NE-SW-trending pre-existing fabrics highly oblique to both the main rift faults and the regional extension direction. The anomalous orientation of the Dilo-Dukana volcanism could be explained by changes in Nubia-Somalia motion (from roughly E-W to NW-SE) during the Quaternary[40]. However, plate kinematic models[41,42], analysis of rift architecture and evolution in the Ethiopian Rift[43,44], as well as the absence of NE-SW trending normal faults in the Ririba rift and surrounding areas do not support this explanation. We suggest instead that a stress reorganisation occurring in response to direct rift linkage, with the local rotation of the stress field to roughly NW-SE in the oblique transfer zone (Fig. 5), may have favoured magma migration along NE-SW-trending pre-existing basement structures. This process may have been enhanced by other factors, such as buoyancy-related stresses[45] due to the difference in topography and crustal thickness between the plateaus and the Turkana depression.

The factors controlling the structural segmentation of continental rifts are still debated[46,47], and the impact of rift segmentation on oceanic transform fault formation is highly controversial[6-8]. Our study suggests that the rift history involves a final locus of focused deformation with an offset geometry dictated by the presence of an inherited transversal heterogeneity[17], underscoring the importance of the pre-rift lithospheric structure on large scale rift architecture and segmentation[48,49]. However, the oblique region connecting the offset Kenyan and Ethiopian rifts lacks transfer faults with significant strike-slip kinematics, indicating that a direct evolution towards an oceanic transform is unlikely[6,7]. Rather, our results show that the major

lateral offset between the Ethiopian and Kenyan rifts in the Turkana depression has evolved from two overlapping, disconnected rifts into a single oblique rift with almost pure extension. We provide structural and temporal evidence that this reorganisation occurred rapidly by abandonment of the lateral tips of the previously overlapping Ethiopian and Kenyan rifts.

Related to this complex evolution, our analysis documents a lack of temporal and structural correlation between recent volcanism and rift-related tectonic activity, with important implications for the rifting process. As in many other rifts, the initiation of rifting in Southern Ethiopia was interpreted, in the absence of other age information, to be marked by the development of Dilo-Dukana volcanic alignments[13–15]. Our data contradict this assumption, and indicate that the age of volcanism cannot necessarily be used as a proxy for rift activity. The other important outcome of the current analysis is that the evolution of deformation in Southern Ethiopia results in volcanic alignments controlled by a complex interplay between local stress field and structural inheritance. This implies that the alignment of volcanic vents may not directly respond to the extension direction imposed by plate motion, suggesting caution in the use of volcanic features as indicators of the regional stress-field[50].

Overall, our results suggest that the process of rift linkage involves phases of rift propagation and abandonment that cause geologically rapid changes (on timescales possibly <1.5 Ma) in the locus and orientation of extension. This type of complexity can be expected to have shaped rifted margins and should be represented in the volcanic and stratigraphic history of proximal margins.

## Methods

**Ar/Ar dating**. Ar/Ar dating experiments were conducted at the dating laboratory of the CNRS-Orléans, France on the groundmass of lava flows selected using stringent criteria for freshness (avoiding scoria and taking the most massive parts). These were processed using standard means (crushing, ultrasonic cleaning, and hand-picking under a binocular microscope) to achieve the highest possible purity. After neutron irradiation at the CLICIT Cd-lined slot of Corvallis Nuclear Reactor (Oregon State University, United States), Ar/Ar experiments were performed using a high-resolution Helix SFT mass-spectrometer outfitted to a home-built $CO_2$-laser based extraction system featuring ultra-low argon blanks. Cold blanks were measured every third gas admission, and include exposure during 6 min to a cold trap (held at −127 °C) and 2 hot GP50 SAES getters. Isotopic abundances were regressed back to inlet time and corrected for instrumental parameters (blank, mass discrimination, dead-time, post irradiation decay and atmospheric contamination) according to ref. [51]. Results of the analysis, summarised in Fig. 3 and Supplementary Table 1 and illustrated in Supplementary Figs. 6a–e and 7, are reported with two different age estimates: the Total Gas Age (TGA) and the Plateau Age (PA). The first one is computed by summing the gas fractions over all steps and the respective error is calculated by quadratic error propagation. The PA is a Maximum Likelihood Estimate (MLE) obtained by inverse-variance averaging of the age (or $^{40}Ar^*/^{39}Ar_K$) of selected steps. PA errors are always lower than the TGA and are usually preferred because MLE allows testing (through the MSWD/(N-1) score, where MSWD is the mean square of weighted deviates) the internal consistency of the pooled data/fractions while allowing to discard conspicuously discordant steps. The PA error estimate is much narrower due to the 1/√N MLE averaging effect. MSWD provides a measure of data scatter relative to analytical precision. It is used to test if an observed distribution fits the theoretical one, taking account of the number of values that are free to vary, namely the degrees of freedom (dof = N-1). Upper and lower cut-off values (UpB and LwB respectively), which take account of the degrees of freedom for each pooled estimate, are reported in Supplementary Table 1, allowing to check if the empirical MDSW/(N-1) value falls in the fiducial range. If this is the case, the scatter in the data can be explained by analytical uncertainties alone. Otherwise, values either <LwB or >UpB would indicate that the analytical uncertainties have been respectively over- or underestimated.

In the diagrams, the heating-steps are represented as boxes, with the width of the box representing the percentage of $^{39}Ar$ released and its thickness corresponding to the analytical error (lower and upper limits of the uncertainty at ±1σ). N refers to the number of steps included in the calculation relative to the total extracted (e.g., N = 20/25 of sample MDZ03 in Supplementary Fig. 6a means that 20 out of a total of 25 steps were included). The step-heating age spectra are plotted along with the evolved $^{37}Ar_{Ca}$, $^{38}Ar_{Cl}$ and $^{36}Ar_{atm}$ normalised to $^{39}Ar_K$ ($^{39}Ar$ produced from $^{39}K$ during irradiation). Tabulated age errors are ±2σ. Decay and isotopic constants used are from ref. [52].

**Chemical analysis of samples of volcanic rocks**. Samples selected for whole-rock analyses were finely grinded in an agata mortar after separation of visible crystal-rich enclaves. Major, minor and trace elements were measured at ALS Minerals Labs (Seville, Spain) with Inductively Coupled Plasma Atomic Emission Spectroscopy for major elements (code ME-ICP06) and Inductively Coupled Plasma Mass Spectrometry for trace elements and REE (codes ME-4ACD81 and ME-MS81D). Accuracy of the analysis was estimated using a set of internal standards (Supplementary Table 3), and comparing the concentration measured for the different elements (or oxides) on each standard with the corresponding median values (±1 standard deviation).

**Analysis of vent alignment**. In order to characterise the distribution of volcanic vents in the area, and to define their relations with faults, we have mapped vent alignments, according to the procedure illustrated in ref. [53]. Vent mapping has been performed on available satellite images and digital elevation models. In particular, vent alignments have been mapped on the basis of the spatial distribution of vents as well as their shapes (vent elongation provides a critical parameter to group single vents into an alignment; Supplementary Fig. 10).

Vent density distribution has been computed by applying a two-dimensional symmetric Gaussian kernel density estimate (ref. [54] and references therein):

$$\lambda(x) = \frac{1}{2\pi N h_i^2} \sum_{i=1}^{N} e^{-\frac{d_i^2}{2h_i^2}} \qquad (1)$$

where $d_i$ is the distance between location $x$ and the $N$ vents, and $h_i$ is the smoothing bandwidth for vent $i$. Distance values between neighbour samples larger than $h_i$ have a small weight in the computation of the density estimate. A variable bandwidth value has been applied by computing the half value of the distance between each sample and its nearest sixth neighbour (ref. [54] and references therein). The vent density map (Supplementary Fig. 10) of the volcanic field has been created by applying Eq. (1), the cells in the resulting matrix have the appropriate vent density value (vent per $km^2$).

The overall shape of the Dilo-Dukana volcanic field has been evaluated applying the Principal Component Analysis (PCA; ref. [54]) to the mapped vents. The PCA analysis provides the direction of the maximum ellipse's axis and the shape of ellipse (Supplementary Fig. 10). The azimuth of the first eigenvalue represents the trend of the long axis of the ellipse that fits the overall trend of the volcanic fields and can be used as a proxy for the field elongation[54]. The PCA has been applied to the Dilo-Dukana volcanic field as well as to the vent clusters in the fields. Notably, clusters show elongation at high angle to the elongation of the volcanic field.

The azimuth distribution of each of the $N(N-1)/2$ segments connecting vent pairs (VVD; Supplementary Fig. 10) has been analysed by constructing the rose diagram and defining the main peak and the azimuthal dispersion. Well-aligned vents will produce a narrow well-defined peak in the azimuth distribution of segments connecting vents and small angular dispersion.

**Analysis of seismicity and seismic moment release**. The earthquake locations and magnitudes used in Figs. 1, 5, and also used to compute the spatial variation in seismic moment release (Fig. 5) are from the National Earthquake Information Center (NEIC) catalogue for the Turkana region and spans 1906 to present (https://earthquake.usgs.gov/earthquakes/search/). We homogenise the magnitudes to the moment magnitude scale of Kanamori[55] producing a catalogue which contains a total of 56 earthquakes with moment magnitudes ranging from 4.3 to 7 $M_w$. We then convert from moment magnitude ($M_w$) to seismic moment release ($N_m$) using the method of Kanamori[55]. To represent the spatial variation in seismic moment release we sum the seismic moment release in grid cells with dimensions of 0.5×0.5 degrees.

**Numerical modelling**. 3D rift evolution has been investigated by using the massively parallel finite element code ASPECT[33–37]. ASPECT's mesh refinement capabilities enable us to focus resolution on the regions of interest (i.e. the near-surface and the region of Kenya/Ethiopia rift interaction), while leaving other areas such as the undisturbed mantle at coarse resolution, leading to kilometre-scale local mesh resolution in 3D.

We employ velocity boundary conditions with a constant extension velocity of 4 mm per year, which corresponds to the long-term average during the last 15 million years (ref. [42]). These kinematic boundary conditions are implemented such that a 2-mm per year velocity is prescribed orthogonal to both model boundaries facing in $x$ direction (Fig. 6). Front and back boundaries feature free slip, the top boundary a free surface. At the bottom boundary, we prescribe a constant vertical inflow of material that balances the outflow through the lateral model sides. The model comprises a domain of 600×1200×165 km in $x$ (cross-rift), $y$ (along-strike) and $z$ direction (depth), respectively. Quadratic shape functions of the finite element method are visualised as piece-wise linear fields at double resolution. This effective resolution varies between 2.5 km in areas of interest (shallower than 60 km depth and for a $y$ coordinate that is in the interval [100 km, 1000 km]), 5 km outside of these areas above 115 km depth, and 10 km beneath 115 km depth.

Temperature boundary conditions feature a constant surface temperature of 0 °C and a bottom temperature of 1350 °C. Lateral boundaries are thermally

isolated. The initial temperature field results from the thermal equilibrium defined by the boundary conditions, the crustal radiogenic heat contribution and the initial depth of the LAB, i.e. the 1350 °C temperature isotherm, which locates at 120 km depth beneath the plateaus, 95 km depth beneath the Turkana depression and 100 km beneath the Ethiopian and Kenyan rifts (Fig. 6). The model accounts for three material layers: felsic crust, mafic crust and mantle (Fig. 6). For the quartz-dominated felsic upper crust, we use the wet quartzite flow law of Rutter and Brodie[56]. The mafic lower crustal layer is represented by a wet anorthite flow law[57] and we use olivine rheology[58] to model deformation of the mantle. All rheological and thermal parameters of these materials are listed in Supplementary Table 4. The initial geometry of the modelled domains and their layer thicknesses are similar to a previous setup[17]; however, in this study we account for the obliquity of the Ethiopian rift, which necessitates a larger model domain. We also employ a significantly higher resolution (locally 2.5 km instead of 7 km), which allows for a more realistic response of the stress field and local extension direction during fault localisation (Fig. 6d). The model involves frictional strain softening, where we linearly reduce the friction coefficient from 0.5 to 0.05 for brittle strain between 0 and 1. For strains >1, it remains constant at 0.05. We also account for viscous strain softening[59] by decreasing the viscosity derived from the ductile flow law by a factor of 2 between viscous strains 0 and 1.

The following model limitations have to be kept in mind when interpreting the results. For reasons of simplicity, we focus on crustal and lithospheric processes and do not account for heterogeneities in the sub-lithospheric mantle[60,61]. Furthermore, we did not include mechanical anisotropies in crust and mantle that might affect localisation processes in nature, especially on the scale of individual faults.

## Code availability

ASPECT is available on GitHub at https://github.com/geodynamics/aspect.

## Data availability

The authors declare that all data supporting the findings of this study are available within the article and its Supplementary Information files, or from the corresponding author upon request.

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

## Acknowledgements

This research was supported by the National Geographic Society (Grant #9976–16, P.I. G. Corti). We thank the DigitalGlobe Foundation for providing the satellite image in Fig. 3. We warmly thank Antonio Zeoli for the processing of the satellite images and Pablo Tierz for valuable discussions. Inversion of fault-slip data and volcanic alignments was obtained using Win-Tensor, a software developed by Dr. Damien Delvaux, Royal Museum for Central Africa, Tervuren, Belgium. D.K. is supported by NERC grant NE/L013932. F.I.-K. is supported by the ECLIPSE Program funded by the New Zealand Ministry of Business, Innovation and Employment. S.B. and A.G. are supported by the Helmholtz Young Investigators Group CRYSTALS (VH-NG-1132). Numerical models were conducted on HLRN cluster Konrad. The Ar/Ar laboratory at ISTO is supported by LABEX Grant "VOLTAIRE".

## Author contributions

G.C. conceived and planned the project, coordinated the interpretation of the data and wrote the paper with contributions from S.B., R.C., D.K., F.I.-K., I.I., F.M., P.M., F.S. and S.S.; S.B., R.C., D.K., F.M. and F.S. contributed to the interpretation of the data. A.E., A.M., R.C., G.C., Z.F., F.S. and S.S. took part to field work and data collection. R.C. supervised the analysis of the volcanic rocks. Z.F. conducted the Ar/Ar dating and the analysis of volcanic rocks. F.S. performed the structural analysis. S.S. supervised the Ar/Ar analysis. P.M. conducted the morphostructural analysis. I.I. and F.M. conducted the statistical analysis of volcanic features. S.B. performed the numerical modelling, with contribution from A.G.; D.K. coordinated the analysis of seismicity, performed by F.I.-K.
