## [Peer Review File · Nature Communications]

Reviewers' comments:

Reviewer #1 (Remarks to the Author):

Summary

Corti et al. present new geological observations and geochronological data from the Ririba rift in southern Ethiopia. Their objective is to understand the spatio-temporal evolution of this little-studied rift and how its life-cycle might link in to the bigger picture of the Ethiopian and Kenyan rifts to the north and south, respectively.

Their new field and structural observations certainly shed light on the architecture of the Ririba rift zone, and they underpin their observations with 5 new Ar-Ar ages and geochemistry. They use their new geochronology, plus some previous K-Ar ages from a nearby volcano (Hurri Hills) to suggest that the Ririba was active for a brief period of only ~1.5 Ma and is now abandoned. At the end of the paper they pull together seismic observations and finite element models to support their observation that a rift tip splay like the Ririba is likely to be abandoned as the larger Ethiopian and Kenyan rift arms connect.

Overall, I enjoyed reading the paper and the authors certainly provide high-quality field and remote sensing observations. In places the manuscript does require quite a bit of flicking back and forth between the main text and supplementary information, but it is worth underscoring that the new insights into the Ririba will be of interest to the rifting community.

I do, however, have several significant concerns regarding the scientific content and presentation of the new results that must be addressed. Firstly, a reader unfamiliar with the region (i.e. most of the Nature Comms readership) would struggle to understand the regional evolution of Turkana which is critical for following the authors' argument. Secondly, I question whether their data really support linkage between the two rift systems today. And finally, the authors need to provide more background to their numerical modelling and clearly demonstrate how it is different and improving on their work published last year (Brune et al., 2017, Tectonics).

These concerns are substantial, but if adequately addressed then I do believe that the topic would be appropriate in scope for Nature Comms. Overall, it is well-written and the author's attempts to combine a variety of observations and models is commendable.

Main comments

1. The overall time-frame of rift evolution in the region is difficult to grasp and not clearly portrayed

The Turkana depression and surrounding rift basins are structurally very complex and there are a large numbers of basins/rifts initiating and abandoning over time. In a short paper it is a real challenge for a reader who is unfamiliar with the region to grasp the wider picture of how these rift basins (e.g., Kenyan, Ethiopian, Chew Bahir, Gofa Province, Kino Sogo, Ririba etc.) are evolving, and what the big picture might be.

While the authors have done a decent job explaining the Ririba, the reader is required to keep all the chronological information in their head. I struggle to understand the regional rifting and volcanism time-frame as well as the uncertainties of the current chronology, and ultimately this leads me to question the validity of their schematics in Fig. 5a-b.

I suggest that they provide a new figure, replacing Figure 4 (which adds little to the manuscript), by a schematic that clearly shows when each rift system and basin in the region was active, and the timing of volcanism too (with age uncertainties shown clearly). I would use a summary schematic like those of Le Turdu et al., 1999 (Palaeogeogr. Palaeoclimatol. Palaeoecol.) and am sure this would turn their rather cryptic description into something that is clear and obvious for the reader to interpret.

A key point to emphasize is the uncertainties on the chronologies. To my understanding the author's suggestion that activity in the Ririba is short (~1.5 Ma), hinges on the K-Ar of the Hurri Hills volcanic samples being 2.3 Ma and not 0.54 Ma. Without the context of these samples and knowing whether they are related to the Hurri flows that cover the Ririba faults I must believe that both ages are equally valid. If the younger age (0.54 Ma) was ascribed, then this would significantly increase Ririba rift timespan by another ~2 Ma. Ultimately, more context for these samples and a clear figure illustrating the regional time-frame is essential.

2. Do the seismic data really support a linkage between the Ethiopian and Kenyan rifts?

One of the key lines of evidence that the Ethiopian and Kenyan rifts are now effectively linked is from seismicity and the authors have calculated the seismic moment release for the region.

However, there is an obvious dearth of tectonic activity in the north of the Kenyan rift over a region of perhaps ~300 km that the authors do not mention.

I think the authors need to address whether this 'seismic gap' is: 1) real, 2) due to a lack of instruments, or 3) statistically-expected based on the recurrence rate of earthquakes in this region. The lack of clarity makes me question their interpretation that the Ethiopian and Kenyan Rifts are truly linked across Lake Turkana today. Could an alternative hypothesis not be that the larger Ethiopian Rift has 'captured' the rift in Lake Turkana – and it now responds to extension in step with Ethiopia, while Kenya is somewhat different?

Clearly rift linkage is a complex process that takes place over millions of years but until the 'seismic gap' is explained I don't believe that it provides overwhelming evidence that these two systems are linked up and responding as one today. A further piece of evidence that the authors could use would be to color volcanoes by different ages so that the reader can more clearly see the temporal change in volcano location and, presumably, the concentration of young volcanoes aligning along the 'linked' rift system?

3. More justification for the modelling, better integration with the geological observations and a clear demonstration of how it differs and builds on the work of Brune et al., 2017 (Tectonics)

The numerical models of rifting are used by the authors to illustrate the spatial pattern of strain evolution as the rifts link. While I do not doubt the quality of their modelling, only one paragraph of the manuscript is dedicated to describing the setup and results, and a small inset to Figure 5 is provided.

My main comment is that the authors should give the model more introduction (why is this numerical model the most appropriate to use?) and clearly describe the parameters that dictate their output geometry. For example, the authors don't even mention in the main text that they impose a pre-existing heterogeneity which will strongly control their model results. The authors also only show a step of the numerical model that appears to match visually with their schematic in Fig. 5b. And it is also not clear whether the timescales of the model are consistent with the geochronology.

While I appreciate that their model will never perfectly predict nature a better presentation of how the model and geological observations might fit is needed. I think they should provide some help for

the reader (in Figure 5) and show what the initial setup is (e.g., something like Fig. S8a) and show a model step that might geologically resemble what is happening in the Pliocene or an earlier phase before rift linkage. In Figure 5d,e the authors label an area that might be applicable to Ririba today – but I think they could go further and also label regions that might resemble Lake Turkana, and the other active and inactive basins.

Finally, because the justification and details of the model setup are so sparse it does make me question whether this work differs substantially from that of Brune et al., 2017 (Tectonics); a paper devoted to modelling deformation and rift evolution in the Turkana region. The authors point out that their new models are run at higher-resolution but to me the results do not look radically different to Brune et al., 2017 (Tectonics, Figures 6 and 7) and again there is no justification of why a higher-resolution is necessary.

One of the differences I note is that Brune et al. (2017) model parallel Ethiopian and Kenyan rifts, whereas this paper models them at an angle. A strength of the Brune et al., 2017 paper is that it predicts the observed westward kink in the Kenyan rift and eastward kink in the Ethiopian rift in Turkana – for me their new models fail to capture these prominent features so well.

I do understand that the authors are trying to present a large-scale picture and are drawing on lots of different methods, but a discussion of the pros and cons of the new and previous numerical models should be considered (at the very least in the supplementary info if there are word count restrictions in the main text)

Minor comments

L. 25 'form'

L. 33-37, reword or split this sentence – not easy to follow and so it lessens the impact of the abstract

L. 51, '...Ririba rift (southern Ethiopia)...'

L. 90-92, repeated sentence

L. 125-127, I don't quite follow the logic here, is it the similar fault orientation, fault architecture, or extension vector that confirms the links to the southern MER, another sentence is needed

L. 140, point to a figure at the end of this sentence?

L. 154, 188 'relatively thin lava platform', here and throughout the manuscript the authors could be more quantitative, 10's, 100's or 1000's of meters?

L. 166, 'poorly variable ages', not clear what you mean, there is limited variation in the ages, i.e. they are tightly clustered?

L. 187, can you label the outcrops of Bulal basalts on Fig. 2? Not clear what you are referring to

L. 194, any photos to support this observation too? It is unfortunate that the lavas don't seem to flow much beyond the fault because its not too clear in Fig. 3

L. 218, suggest you remove ', therefore arguing', replace with 'which argues against'

L. 271-272, what do you mean by abundant, are they found in every outcrop, or just the samples you have studied?

L.278-281, I like the idea that geochemistry 'might' provides a method for fingerprinting changes in stress regime in these regions – perhaps there are wider implications for interpreting deformation in ancient rift zones?

L. 297, 'highly three dimensional' from my reading this is the first time this idea is mentioned, I presume it is linked to the findings of the modelling and the need for lithospheric heterogeneities, but this idea has not really come through clearly in the paper. I think more justification and detail on the modelling would clarify this. Also 'rapid changes', again I would like to quantify, ka, Ma etc.?

L. 411, why is a 'higher resolution' needed for this study compared to the previous work in ref. 17?

Figure 1

Double check all relevant basins labelled – Segen Basin is missing for example, Chamo basin mentioned by Ebinger et al., 2002 (GSAB)

No Quaternary volcanoes in the MER are labelled – consult Smithsonian Global Volcanism database or other regional overviews, e.g. Le Turdu et al., 1999 (Palaeogeogr. Palaeoclimatol. Palaeoecol.); Fontijn et al., 2018 (JVGR).

Caption – I'm not entirely clear on the term 'deformation domains', is there sound quantitative evidence that these zones are all deforming in unison or is it qualitative, i.e. faults seem to align, a reference and or brief explanation would suffice

Figure 2

I presume the samples dated from Hurri Hills volcano are from further south and not shown? Still it would be helpful to have the ages and an arrow pointing south.

For normal faults – how much throw in meters, approximately, are we talking about for each category?

Figure 3

Can the new age data not be added to this plot?

Perhaps a few arrows to show lava flow direction, would be helpful. And to clearly show the reader that they spill over the faults.

Figure 4

As mentioned above I don't think this Figure adds much to the manuscript and could be removed (or placed with other Ar-Ar results in the Supplementary Info)

Figure 5

Coloring the volcanoes by age may help emphasize progression in volcanism

d,e more could be done to integrate the modelling results and link them to the geological observations – see my main comment

if you do have an inset (e.g., e) then you need to show which area of d this corresponds to

Need to confirm that model scale is analogous to real observations too

Supplementary information

Section 1

The sentence starting 'The D/L relations...' is too long and difficult to follow

'This supports that...' change to 'This supports our idea that...' or similar

Section 2

First sentence, 'two different age estimates'

Supplementary Table 1, decimal points rather than commas, 'no data' instead of ?

Ar-Ar ages, I could be wrong, but I would have thought standard practice of geochronology is to report 2 sigma?

Section 3

Can you provide a basic assessment of the accuracy and precision of the geochemical measurements, I would normally expect a statement of this, or inclusion of standard data

2nd paragraph, replace 'representing' with 'from', again it would be helpful to quantify how much you mean by abundant, 1, 10, 50 %?

Supplementary Table 2, more details on sample location and rock type needed, i.e., standard procedure that will allow other authors to use this information

Section 4

Fig. S5, location inset and scale bar for d-f is needed

Section 5

Usage of the word 'evidences' is incorrect, 'There are multiple lines of evidence ...'

Section 6

2nd paragraph, 'Apart three...', correct

Section 7

See main comment – I think lots more information comparing this work and Brune et al., 2017 (Tectonics) should be included here to clearly distinguish the two bodies of work

Fig. S8a, can the scale not be in km rather than m? Is there a timescale element that needs to be included in these plots too?

2nd paragraph, the author's have tried some integration with geological observations by labelling A-C in Fig. S9 but I think more could be done to help the reader. For example, are A and B supposed to be analogous to the active stage of the Ririba, while C is the inactive stage? This is not obvious in the text.

Reviewer #2 (Remarks to the Author):

This manuscript presents new structural and chronological constraints on a relatively poorly studied rift basin in the East Africa Rift System in the Turkana depression. Corti and coauthors are able to document both the amount of extension, the geometry and displacement of rift faults and volcanic centers, and the timing of rift propagation in this region. They use these new constraints together with published studies from other parts of the Turkana depression to develop a new model for rift segment linkage. They show that segment linkage and resulting changes to rift system happen very quickly, particularly considering the slow extension rates here. As linkage occurs, a new continuous rift zone forms without transfer zones, and in which stress re-orientation has occurred such that pure dip-slip motion is occurring. Volcanism continued in the Ririba rift region even after it appears to have been abandoned. The results are very interesting and relevant to rift systems worldwide, and thus is appropriate for publication in Nature Communications. Overall, I found the paper to be well-written and illustrated and convincing. I have a few comments and questions below, which I think can be addressed with minor revisions.

1. My primary comments it that the discussion and illustration of the role of pre-existing structures on rifting needs to be improved to be clearer.

-The primary pre-existing structure described in the text is a NW-SE oriented Mesozoic basin that cross cuts this region. However, pre-existing structures are evoked to explain many other observations, including the displacement profiles and orientations of rift faults in the Ririba rift (which are dominantly N-S) and the orientation of Quaternary volcanic centers (which are dominantly NE-SE). More information needs to be given on the evidence for other types of pre-existing structures besides the NW-SE oriented Mesozoic basin that may have influenced the faults and volcanic centers.

-The authors describe the complex range of orientations of rift faults and the displacement profiles on those faults as evidence that rift faults are strongly influence by pre-existing structures. However, I was struck by the overall uniformity of orientations of faults in the rose diagram shown in Figure 2c, and the fact that the dominant N-S orientation is ideal for the E-W direction of extension in this region. The main two faults (F1 and F2) for which displacement profiles were calculated in the supplement are also N-S oriented faults (not a complex, non-ideal orientation). Are there N-S oriented pre-existing structures that are ideally oriented for the modern extension direction or do these faults actually cross-cut preexisting structures?

-A number of factors are proposed to controls the orientation of the Quaternary volcanic centers. Early in the manuscript, the orientations are attributed to preexisting structures (Lines 171-180), but later a wide-range of factors are evoked, including the modified stress field, changing crustal structure (Lines 277-281). This should be clarified.

-Given the importance of pre-existing structures to the authors' model for rift development, I think it would make sense to represent what is known about the location and orientation of these features on one of the maps so they can be compared with the faults and volcanic centers. In particular, the Mesozoic rift basin, the Buluk Fault zone and any N-S oriented pre-existing structures that may have been exploited by Ririba rift faults as implied by text.

2. The decoupling between the timing of extension and magmatism is very interesting and important. I think this should be emphasized more in the Broader implications. This is very important for understanding other more mature rift systems. In the absence of other age information, the age of volcanism is sometimes used as a proxy for rift activity.

3. The snapshots of dynamical models shown in the main text do not feel very well connected to the observations presented in the paper since they show later stage of extension after the abandonment of the Ririba Rift. I suggest replacing one of the panels in Figure 5 with one of the panels from Suppl Fig 9 that shows the earlier, broader zone of deformation. One interesting difference between the numerical models and observations is that during the broad deformation stage, there appear to be many closely spaced smaller faults in the model compared with three relatively localized rift zones in the observations. What explains this difference?

Other minor comments:

Lines 90-92 – Repeated sentence.

Line 116 – Give reference for regional plate kinematics

Line 278 – What is meant by “a reduction in plate motion related stresses”?

Figure 1 – add location of zoomed map to inset in lower right hand corner

-Suppl Fig 1 - It would be nice to plot the displacement profiles for F1 and F2 at the same scale and align them spatially so that its easier to compare patterns of displacement between the two. Was a similar analysis done on other faults with less ideal orientation for comparison?

-Suppl Fig 9 – What are A, B and C labeled in these panels?

The detailed responses to the Reviewers' are coded as follows:

Reviewer's comments: *italics (11pt)*

Responses: plain text (13 pt)

Reviewer #1 (Remarks to the Author):

Summary

Corti et al. present new geological observations and geochronological data from the Ririba rift in southern Ethiopia. Their objective is to understand the spatio-temporal evolution of this little-studied rift and how its life-cycle might link in to the bigger picture of the Ethiopian and Kenyan rifts to the north and south, respectively.

Their new field and structural observations certainly shed light on the architecture of the Ririba rift zone, and they underpin their observations with 5 new Ar-Ar ages and geochemistry. They use their new geochronology, plus some previous K-Ar ages from a nearby volcano (Hurri Hills) to suggest that the Ririba was active for a brief period of only ~1.5 Ma and is now abandoned. At the end of the paper they pull together seismic observations and finite element models to support their observation that a rift tip splay like the Ririba is likely to be abandoned as the larger Ethiopian and Kenyan rift arms connect.

Overall, I enjoyed reading the paper and the authors certainly provide high-quality field and remote sensing observations. In places the manuscript does require quite a bit of flicking back and forth between the main text and supplementary information, but it is worth underscoring that the new insights into the Ririba will be of interest to the rifting community.

I do, however, have several significant concerns regarding the scientific content and presentation of the new results that must be addressed. Firstly, a reader unfamiliar with the region (i.e. most of the Nature Comms readership) would struggle to understand the regional evolution of Turkana which is critical for following the authors' argument. Secondly, I question whether their data really support linkage between the two rift systems today. And finally, the authors need to provide more background to their numerical modelling and clearly demonstrate how it is different and improving on their work published last year (Brune et al., 2017, Tectonics).

These concerns are substantial, but if adequately addressed then I do believe that the topic would be appropriate in scope for Nature Comms. Overall, it is well-written and the author's attempts to combine a variety of observations and models is commendable.

We really thank the Reviewer. We believe the manuscript has really benefited from his/her comments, which were really pertinent and constructive!

Main comments

1. The overall time-frame of rift evolution in the region is difficult to grasp and not clearly portrayed

The Turkana depression and surrounding rift basins are structurally very complex and there are a large numbers of basins/rifts initiating and abandoning over time. In a short paper it is a real challenge for a reader who is unfamiliar with the region to grasp the wider picture of how these rift basins (e.g., Kenyan, Ethiopian, Chew Bahir, Gofa Province, Kino Sogo, Ririba etc.) are evolving, and what the big picture might be.

While the authors have done a decent job explaining the Ririba, the reader is required to keep all the chronological information in their head. I struggle to understand the regional rifting and volcanism time-frame as well as the uncertainties of the current chronology, and ultimately this leads me to question the validity of their schematics in Fig. 5a-b.

We agree with the Reviewer and we have modified the manuscript accordingly to better introduce the tectonics of the Turkana depression and the evolution of the many different basins composing it.

In particular, we have modified the main text and referred the interested reader to a new section in the Supplementary material (Supplementary Section 1) where we now treat in detail the Oligocene-Recent evolution of the region.

In this section, we have also added two figures (Supplementary Figs 1 and 2) to graphically illustrate the volcano-tectonic evolution of the Turkana depression. Specifically, in new Supplementary Figure 1 we show a chronogram of the spatial and temporal evolution of the volcano-tectonic activity based on a compilation of available data on the many basins composing the Turkana depression and surrounding regions. This was a rather difficult task to be achieved, but was facilitated by a recent collection of data by Boone et al, Tectonics, which we integrated in different areas with data from other works – especially concerning the volcanic activity.

The other figure we added (Supplementary Figure 2) is a schematic representation of the previous knowledge on the Pliocene-Recent rift propagation in the Turkana-Ririba area (something that the reader can compare to our new interpretation illustrated in Fig. 5, main text). Also reported in the figure is a chronogram similar to that present in Supplementary Figure 1 but restricted to the area between latitudes 3°N and 4°N, which corresponds to the Turkana depression *sensu stricto*. This is used to clearly show the E-W migration of deformation from the Oligocene to Recent times, as can be deduced from previous papers.

[note: we think both figures can be placed in the supplementary material since - although important to clarify the evolution of the region- they are not really central to our analysis of the Ririba rift. Of course, both figures (or maybe Supplementary Figure 1) can be moved to the main text of requested by the Reviewer and/or the Editor.]

I suggest that they provide a new figure, replacing Figure 4 (which adds little to the manuscript), by a schematic that clearly shows when each rift system and basin in the region was active, and the timing of volcanism too (with age uncertainties shown clearly). I would use a summary schematic like those of Le Turdu et al., 1999 (Palaeogeogr. Palaeoclimatol. Palaeoecol.) and am sure this

would turn their rather cryptic description into something that is clear and obvious for the reader to interpret.

This is a very good suggestion. We followed it by introducing in the main text a new final figure (new Fig. 7), which clearly illustrates the spatial and temporal evolution of the volcano-tectonic evolution of the Ririba rift and surrounding regions, according to our new data (as suggested by the Reviewers age uncertainties are shown in the Figure).

The style we have chosen is somehow comparable to the one in Le Turdu et al., and we think this is a very efficient way to compare our new observations with the results of numerical modeling, in terms of location and lateral extent of areas affected by tectonic activity during rift evolution (note the very good correspondence between observations and model predictions). Moreover, the style of the chronogram in this new Fig 7 is the same as the one we used in new Supplementary Figure 2, so that the reader can easily compare the previous interpretation of the evolution of the region with our new findings.

A key point to emphasize is the uncertainties on the chronologies. To my understanding the author's suggestion that activity in the Ririba is short (~1.5 Ma), hinges on the K-Ar of the Hurri Hills volcanic samples being 2.3 Ma and not 0.54 Ma. Without the context of these samples and knowing whether they are related to the Hurri flows that cover the Ririba faults I must believe that both ages are equally valid. If the younger age (0.54 Ma) was ascribed, then this would significantly increase Ririba rift timespan by another ~2 Ma. Ultimately, more context for these samples and a clear figure illustrating the regional time-frame is essential.

Unfortunately, we could not sample the Huri Hills during our two fieldworks, because of logistic problems (e.g., difficulty to access the area) and (more importantly) because we had permission to work and collect samples in Ethiopia only and we could not cross the border to Kenya (where the volcanic edifice is almost entirely located). Therefore, we could not provide detailed geochronological constraints on the activity of this large shield volcano and we had to rely on available data.

We have now reviewed this information in more detail, and had access to the geological map of the North Horr region by Charsley (1988) (which includes the Hurri Hills), and all these data reinforce our previous analysis, as summarized below.

The only published data on the Hurri Hills activity are the K-Ar ages by Brotzu et al. (1984) and those reported in the geological map of the North Horr (Charsley, 1988). According to these ages and to the geological map and explanatory notes of Charsley (1988), the present edifice is formed by a Late Pliocene basal lava pile, forming a large shield volcano, overlain by the products (lava flows and tephra blankets) of a large number of Pleistocene monogenetic vents (mainly cinder cones) located on the shield surface.

The basal flood lavas of the edifice started emplacing at around 3 Ma, with a sample from the southwestern slope of the edifice giving a K-Ar age of 2.8 Ma (see new Supplementary Figure 7). This timing is consistent with these lavas overlying the Bulal basalts, which we dated at ca 3.6-3.7 Ma in the area of Dilo. Conversely, the lava flows and tephra products of the following monogenetic activity span a quite large time period, from an oldest age of 2.3 Ma (K-Ar age of Brotzu et al., 1984) up to a mid-Pleistocene age (K-Ar age of 0.54 Ma reported in Charsley, 1988) for some products of summit vents.

Satellite images clearly show that the lavas of the Hurri Hills shield seal the Ririba faults at the northern termination of the shield. Therefore, since the oldest reported age for the shield volcano lavas is 2.8 Ma, and the shield is overlain by younger products with ages as old as 2.3 Ma, the end of rift activity likely occurred in the time interval between 2.8 and 2.3 Ma. Of course, the possibility exists that the lavas covering the faults may be even older than 2.8 Ma (e.g., the ca. 3 Ma age reported in Charsley, 1988), therefore further reducing the duration of faulting in the Ririba Rift. Future work is needed to decrease the uncertainty in the timing of rift deactivation.

[it is important to emphasize that the uncertainty in the timing of activity is not decreasing the impact of our findings, since the most important conclusion of our analysis is the documented inactivity of the Ririba Rift and the migration of deformation to nearby regions.]

Anyway, following the Reviewer's comment, we have now included a more detailed explanation for this in a new supplementary section (Supplementary Section 6), we have modified Fig 2 and introduced a new supplementary figure (Supplementary Figure 7) to better illustrate the chronological data in the area of interest.

Importantly, as suggested by the Reviewer, we have now introduced a detailed chronogram to illustrate the spatial and temporal evolution of the volcanic and tectonic activity (including uncertainties) and compare these findings with numerical predictions in new Fig. 7. We agree with the Reviewer in that this new figure was essential to clearly illustrate our new findings and their implications for rift evolution in the region.

2. Do the seismic data really support a linkage between the Ethiopian and Kenyan rifts?

One of the key lines of evidence that the Ethiopian and Kenyan rifts are now effectively linked is from seismicity and the authors have calculated the seismic moment release for the region. However, there is an obvious dearth of tectonic activity in the north of the Kenyan rift over a region of perhaps ~300 km that the authors do not mention.

I think the authors need to address whether this 'seismic gap' is: 1) real, 2) due to a lack of instruments, or 3) statistically-expected based on the recurrence rate of earthquakes in this region.

The lack of clarity makes me question their interpretation that the Ethiopian and Kenyan Rifts are truly linked across Lake Turkana today. Could an alternative hypothesis not be that the larger Ethiopian Rift has 'captured' the rift in Lake Turkana – and it now responds to extension in step with Ethiopia, while Kenya is somewhat different?

Clearly rift linkage is a complex process that takes place over millions of years but until the 'seismic gap' is explained I don't believe that it provides overwhelming evidence that these two systems are linked up and responding as one today.

The seismic gap in the north of the Kenyan rift is likely due to a combination of lack of instruments in the region and is also statistically expected based on the recurrence rate of earthquakes in the region. The presence of a linear chain of volcanoes along the rift axis suggests that a large proportion of strain in the northern Kenyan rift is taken up by magma intrusion, much like the northern Ethiopian rift. Seismic strain is likely reduced as a proportion of total strain, making sampling the full seismic cycle impossible with a several decade long catalogue. In addition, the presence of magmatism likely shallows the brittle – ductile transition, thus reducing the maximum magnitude of earthquake, making sampling the seismicity with a global catalogue of magnitude of completeness around M4 also

difficult. We have found results from local/regional seismicity experiment conducted in central and northern Kenya from 1981 (Pointing et al., 1985) in which 2000 micro seismic events of magnitudes less than 3.1 are located with location error bars of less than 5km. Seismicity in northern Kenya is concentrated along the Suguta Valley and Lake Turkana (see figure below), supporting our interpretation. Reference to this work has been added in the main text.

Our interpretation of the current zone of strain is also supported by the interpreted plate boundary in plate kinematic models derived from GPS data, combined with geological evidence such as position of young volcanoes, and the new constraints on timing of fault activity presented in our manuscript.

Figure 1 Revisions. Seismicity of northern Kenya for the period January to August 1981 reported in Pointing et al. (1985). Seismic stations are squares and labelled with a three letter code. Earthquakes are circles and crosses.

Reference

Pointing , A.J., Maguire, P.K.H., Khan, M.A., Francis, D.J., Swain, C.J., Shah, E.R., Griffiths, D.H. (1985) Seismicity of the northern part of the Kenya rift valley, *Journal of Geodynamics* 3, 23-37.

A further piece of evidence that the authors could use would be to color volcanoes by different ages so that the reader can more clearly see the temporal change in volcano location and, presumably, the concentration of young volcanoes aligning along the 'linked' rift system?

This is a very good suggestion. We have now modified the figure by differentiating the volcanic edifices according to the timing of their activity. In particular, we now differentiate edifices with activity in the Pleistocene, Holocene, Holocene-uncertain; note that it is very difficult to illustrate pre-Pleistocene activity since in many cases it consists of widespread fissural eruptions not directly linked to individual edifices (see for instance the Bulal basalts characterizing the Ririba rift and surrounding regions).

Anyway, this new representation reinforces our interpretation of a focused volcano-tectonic activity in a continuous deformation corridor linking the two rift valleys, although with some complications (e.g., the Mega volcanic field, which is unrelated to rifting as we clearly explain in the text).

3. More justification for the modelling, better integration with the geological observations and a clear demonstration of how it differs and builds on the work of Brune et al., 2017 (Tectonics)

The numerical models of rifting are used by the authors to illustrate the spatial pattern of strain evolution as the rifts link. While I do not doubt the quality of their modelling, only one paragraph of the manuscript is dedicated to describing the setup and results, and a small inset to Figure 5 is provided.

My main comment is that the authors should give the model more introduction (why is this numerical model the most appropriate to use?) and clearly describe the parameters that dictate their output geometry. For example, the authors don't even mention in the main text that they impose a pre-existing heterogeneity which will strongly control their model results. The authors also only show a step of the numerical model that appears to match visually with their schematic in Fig. 5b. And it is also not clear whether the timescales of the model are consistent with the geochronology.

We fully agree. Following this comment, we have now modified the ms to explain better why we use a new numerical modelling approach, we now explain in more detail the numerical modelling results, and what are the main differences with our previous modelling work (Brune et al., 2017). In a nutshell, the new model allows us to zoom into regions of interest such as the Ririba rift and to extract local extension direction, which was not possible with the same detail before.

While I appreciate that their model will never perfectly predict nature a better presentation of how the model and geological observations might fit is needed. I think they should provide some help for the reader (in Figure 5) and show what the initial setup is (e.g., something like Fig. S8a) and show a model step that might geologically resemble what is happening in the Pliocene or an earlier phase before rift linkage.

Following this comment (and a similar comment by Reviewer 2, see below) we have now modified the manuscript to better illustrate the setup/evolution of the numerical model and how its results fit with the evolution of rift linkage in nature. In particular, we have followed the Reviewer's recommendations and added the initial model set-up and two additional stages of evolution in the main text. The easiest way to do this, was to move the illustration

of numerical modeling to a new figure (Figure 6), separating it from the evolutionary sketch derived from geological data (Fig. 5).

Importantly, as stated above, we have now introduced a new figure (Fig. 7), which integrates data on the activity of the Ririba rift and its surroundings with numerical model results. There is a very good first-order fit between modeling predictions and natural observations, with the main steps of deformation localization in the models roughly corresponding to those characterizing nature. The differences between models and observations in Fig.7 might result from the role of inherited anisotropies in nature that are not captured by the model, but we also discuss an alternative speculation that the diffuse deformation at the onset of rifting in the Ririba area has simply not yet been discovered in the field.

In Figure 5d,e the authors label an area that might be applicable to Ririba today – but I think they could go further and also label regions that might resemble Lake Turkana, and the other active and inactive basins. Thank you for that suggestions, we have now labelled other regions of interest in panel d of new Figure 6.

Finally, because the justification and details of the model setup are so sparse it does make me question whether this work differs substantially from that of Brune et al., 2017 (Tectonics); a paper devoted to modelling deformation and rift evolution in the Turkana region. The authors point out that their new models are run at higher-resolution but to me the results do not look radically different to Brune et al., 2017 (Tectonics, Figures 6 and 7) and again there is no justification of why a higher-resolution is necessary. One of the differences I note is that Brune et al. (2017) model parallel Ethiopian and Kenyan rifts, whereas this paper models them at an angle. A strength of the Brune et al., 2017 paper is that it predicts the observed westward kink in the Kenyan rift and eastward kink in the Ethiopian rift in Turkana – for me their new models fail to capture these prominent features so well.

I do understand that the authors are trying to present a large-scale picture and are drawing on lots of different methods, but a discussion of the pros and cons of the new and previous numerical models should be considered (at the very least in the supplementary info if there are word count restrictions in the main text)

We significantly extended the description of the model setup and results in the main text. We also explicitly isolate the differences between Brune et al. 2017 and this study. The main differences are that we have an oblique angle in the Ethiopian rift, which necessitates a larger model domain and that the higher resolution in the new model allows a more detailed small-scale kinematic analysis of the computed results (see discussion section). In order to fit the observational temporal constraints with the model evolution, we had to change model parameters from the original setup in the Brune et al. 2017 study, most importantly the strain softening configuration. These parameters are difficult to constrain from observation or laboratory experiments, which is why we used them to gain a better fit to observations. Although the westward kink of the Kenyan rift is not very well pronounced, the eastward kink of the Ethiopian rift, despite being less prominent than in the previous models, is very well visible in Fig. 6 and Supplementary Fig. 14 (where parts of it are labelled as Ririba rift). It corresponds indeed to southernmost tip of the Ethiopian rift, which overlaps with the tip of the Kenya rift in the Turkana depression and is progressively abandoned during rift evolution (see Figs.). We are therefore convinced that the model scenario very well reproduces the first-order evolution of rift linkage and localization in the study region (see also new Fig. 7).

Minor comments

L. 25 'form'

Done

L. 33-37, reword or split this sentence – not easy to follow and so it lessens the impact of the abstract

Done

L. 51, '...Ririba rift (southern Ethiopia)...'

Done

L. 90-92, repeated sentence

Text modified

L. 125-127, I don't quite follow the logic here, is it the similar fault orientation, fault architecture, or extension vector that confirms the links to the southern MER, another sentence is needed

Text modified to clarify this.

L. 140, point to a figure at the end of this sentence?

Text modified accordingly

L. 154, 188 'relatively thin lava platform', here and throughout the manuscript the authors could be more quantitative, 10's, 100's or 1000's of meters?

We have now indicated the thickness of the lava platform (as reported in Hackman et al., 1990 and Vetel and LeGall, 2006) in section "Volcanic activity"

L. 166, 'poorly variable ages', not clear what you mean, there is limited variation in the ages, i.e. they are tightly clustered?

Text modified accordingly

L. 187, can you label the outcrops of Bulal basalts on Fig. 2? Not clear what you are referring to

We agree in that this was maybe not clear in the previous version of Fig.2. We have now modified it and indicated the Bulal basalts in the figure legend.

L. 194, any photos to support this observation too? It is unfortunate that the lavas don't seem to flow much beyond the fault because its not too clear in Fig. 3

Unfortunately, we do not have field photos of this, but the high-resolution DigitalGlobe satellite images clearly show this (along with many other geomorphological evidences, summarized in the main text and in Supplementary Section 6). To better illustrate this we have added to the Supplementary material a new figure (new Supplementary Fig. 11), which zooms to point A in Fig. 3.

Following this comment, we realized that we did not show field photos of the Ririba rift; we now show a photo of the main boundary faults in a new figure (Supplementary Figure 3).

L. 218, suggest you remove ' , therefore arguing', replace with 'which argues against'

Text modified accordingly

L. 271-272, what do you mean by abundant, are they found in every outcrop, or just the samples you have studied?

Mantle xenoliths are very abundant in all the volcanic products outcropping in the area. We have modified the text to clarify this

L.278-281, I like the idea that geochemistry ‘might’ provides a method for fingerprinting changes in stress regime in these regions – perhaps there are wider implications for interpreting deformation in ancient rift zones?

We agree and we have added a sentence to describe this in the Discussion section

L. 297, ‘highly three dimensional’ from my reading this is the first time this idea is mentioned, I presume it is linked to the findings of the modelling and the need for lithospheric heterogeneities, but this idea has not really come through clearly in the paper. I think more justification and detail on the modelling would clarify this.

We agree with the Reviewer in that using the term 'three-dimensional' here was not clear. We have now rearranged the sentence to clarify our concepts.

Also ‘rapid changes’, again I would like to quantify, ka, Ma etc.?
Text rearranged accordingly

L. 411, why is a ‘higher resolution’ needed for this study compared to the previous work in ref. 17?

One of the main aims of these high-resolution models was to focus on details that the previous models (Brune et al., 2017) were unable to capture. For instance, the models were designed to test the hypothesis of a stress field reorientation following rift linkage in the region. The variations we observe in the current models are not large enough to be captured by the previous, lower-resolution models.

Figure 1

Double check all relevant basins labelled – Segen Basin is missing for example, Chamo basin mentioned by Ebinger et al., 2002 (GSAB)

Segen basin added. All the other relevant basins should be included now.

No Quaternary volcanoes in the MER are labelled – consult Smithsonian Global Volcanism database or other regional overviews, e.g. Le Turdu et al., 1999 (Palaeogeogr. Palaeoclimatol. Palaeoecol.); Fontijn et al., 2018 (JVGR).

Good point. Figure modified.

Caption – I’m not entirely clear on the term ‘deformation domains’, is there sound quantitative evidence that these zones are all deforming in unison or is it qualitative, i.e. faults seem to align, a reference and or brief explanation would suffice.

The deformation zones are identified in previous works, to which we refer now in the manuscript. It is more a qualitative definition, which is also based on the overall evolution of the region.

Figure 2

I presume the samples dated from Hurri Hills volcano are from further south and not shown? Still it would be helpful to have the ages and an arrow pointing south.

We have now indicated the location of samples from the Hurri Hills (Brotzu et al., 1984) in a new Supplementary Figure (Supplementary Figure 7).

For normal faults – how much throw in meters, approximately, are we talking about for each category?

Figure legend modified accordingly

Figure 3

Can the new age data not be added to this plot?

We prefer not to add the data to this figure, but we have added all the new (and existing) ages in a new Supplementary Figure (Supplementary Figure 7), which also includes samples from the Hurri Hills (as explained above). Anyway, if the Reviewer (or the Editor) thinks that modifying Figure 3 to include the new geochronological information is strictly necessary, we can modify it to add the data.

Perhaps a few arrows to show lava flow direction, would be helpful. And to clearly show the reader that they spill over the faults.

Good point. Figure modified accordingly.

Figure 4

As mentioned above I don't think this Figure adds much to the manuscript and could be removed (or placed with other Ar-Ar results in the Supplementary Info)

We agree with the Reviewer but we also think that this is a good summary of the geochronological results. So, since there is no space problem with this manuscript, we prefer to keep this figure in the main text. However, if the Editor thinks that we should move it to the Supplementary material, no problem and we will move it.

Figure 5

Coloring the volcanoes by age may help emphasize progression in volcanism

d,e more could be done to integrate the modelling results and link them to the geological observations – see my main comment

if you do have an inset (e.g., e) then you need to show which area of d this corresponds to

Need to confirm that model scale is analogous to real observations too

We have now labeled the volcanoes according to the different timing of activity. And, as stated above, we have also added a new figure (Fig. 7) to better show the comparison between models and nature

Supplementary information

Section 1

The sentence starting 'The D/L relations...' is too long and difficult to follow

Sentence modified accordingly.

'This supports that...' change to 'This supports our idea that...' or similar

Done

Section 2

First sentence, 'two different age estimates'

Done

Supplementary Table 1, decimal points rather than commas, 'no data' instead of?

Table modified accordingly

Ar-Ar ages, I could be wrong, but I would have thought standard practice of geochronology is to report 2 sigma?

Table modified accordingly

Section 3

Can you provide a basic assessment of the accuracy and precision of the geochemical measurements, I would normally expect a statement of this, or inclusion of standard data

Ok, we have better specified the methods in the Supplementary material and added a table with the measures of the standards (Suppl Table 3)

2nd paragraph, replace 'representing' with 'from',

Done

again it would be helpful to quantify how much you mean by abundant, 1, 10, 50 %?

Text modified accordingly

Supplementary Table 2, more details on sample location and rock type needed, i.e., standard procedure that will allow other authors to use this information

Good suggestion. table modified accordingly.

Section 4

Fig. S5, location inset and scale bar for d-f is needed

Figure modified accordingly

Section 5

Usage of the word 'evidences' is incorrect, 'There are multiple lines of evidence ...'

Text modified accordingly.

Section 6

2nd paragraph, 'Apart three...', correct

Done

Section 7

See main comment – I think lots more information comparing this work and Brune et al., 2017 (Tectonics) should be included here to clearly distinguish the two bodies of work

As discussed above, we added a comparison to the previous study in the main manuscript.

Fig. S8a, can the scale not be in km rather than m? Is there a timescale element that needs to be included in these plots too?

Good point. We changed the scales in Fig 6 and Supplementary Fig. 14 and we converted extension to times.

2nd paragraph, the author's have tried some integration with geological observations by labelling A-C in Fig. S9 but I think more could be done to help the reader. For example, are A and B supposed to be analogous to the active stage of the Ririba, while C is the inactive stage? This is not obvious in the text.

We added the according labels to Fig 6 and Supplementary Fig. 14

Reviewer #2 (Remarks to the Author):

This manuscript presents new structural and chronological constraints on a relatively poorly studied rift basin in the East Africa Rift System in the Turkana depression. Corti and coauthors are able to document both the amount of extension, the geometry and displacement of rift faults and volcanic centers, and the timing of rift propagation in this region. They use these new constraints together with published studies from other parts of the Turkana depression to develop a new model for rift segment linkage. They show that segment linkage and resulting changes to rift system happen very quickly, particularly considering the slow extension rates here. As linkage occurs, a new continuous rift zone forms without transfer zones, and in which stress re-orientation has occurred such that pure dip-slip motion is occurring. Volcanism continued in the Ririba rift region even after it appears to have been abandoned. The results are very interesting and relevant to rift systems worldwide, and thus is appropriate for publication in Nature Communications. Overall, I found the paper to be well-written and illustrated and convincing. I have a few comments and questions below, which I think can be addressed with minor revisions.

1. My primary comments is that the discussion and illustration of the role of pre-existing structures on rifting needs to be improved to be clearer.

We agree with reviewer on this point and we have improved the manuscript accordingly, see below.

-The primary pre-existing structure described in the text is a NW-SE oriented Mesozoic basin that cross cuts this region. However, pre-existing structures are evoked to explain many other observations, including the displacement profiles and orientations of rift faults in the Ririba rift (which are dominantly N-S) and the orientation of Quaternary volcanic centers (which are dominantly NE-SE). More information needs to be given on the evidence for other types of pre-existing structures besides the NW-SE oriented Mesozoic basin that may have influenced the faults and volcanic centers.

According to this comment, we have now modified the manuscript and explained in more detail in Supplementary Section 2 the different basement fabrics characterizing the area, and the relations with faults (and volcanic features) in the area. We have now introduced a new figure (Supplementary Figure 5) to illustrate this.

-The authors describe the complex range of orientations of rift faults and the displacement profiles on those faults as evidence that rift faults are strongly influence by pre-existing structures. However, I was struck by the overall uniformity of orientations of faults in the rose diagram shown in Figure 2c, and the fact that the dominant N-S orientation is ideal for the E-W direction of extension in this region. The main two faults (F1 and F2) for which displacement profiles were calculated in the supplement are also N-S oriented faults (not a complex, non-ideal orientation). Are there N-S oriented pre-existing structures that are ideally oriented for the modern extension direction or do these faults actually cross-cut preexisting structures?

The main faults of the Ririba rift (e.g., F1, F2) are oriented roughly NNW-SSE (N170°E), i.e., they are not orthogonal to the direction of extension (roughly N100°E) in the area; their trend deviates of about 20° with respect to the ideal, extension-orthogonal direction.

Conversely, the main boundary faults are parallel to NNW-SSE basement fabrics, which diffusely characterize the area (see for instance Vetel et al., 2005) and also control the orientation of the main boundary faults of the Chow Bahir basin (e.g., Corti, 2009; Philippon et al., 2014).

Despite being dominated by the long boundary faults, the Ririba rift and surrounding areas are characterised by complex sets of variably oriented faults, which give rise to minor peaks in the rose diagram of fault distribution of Fig. 2c. Analogously to the boundary faults, these different sets of faults are not orthogonal to extension, and give rise to complex angular patterns that parallel the NNE-SSW, NE-SW or ENE-WSW basement trends.

This, along with the analysis of the displacement/length ratio and the shape of the displacement/length curve, supports a strong influence of the pre-existing structures on fault development in the Ririba rift.

This is now more clearly stated in the main text and explained in detail in Supplementary Section 2.

-A number of factors are proposed to controls the orientation of the Quaternary volcanic centers. Early in the manuscript, the orientations are attributed to preexisting structures (Lines 171-180), but later a wide-range of factors are evoked, including the modified stress field, changing crustal structure (Lines 277-281). This should be clarified.

We agree with the Reviewer in that the sentence related to the stress field changes was not clear and we have now rearranged the main text to clarify these concepts.

-Given the importance of pre-existing structures to the authors' model for rift development, I think it would make sense to represent what is known about the location and orientation of these figures on one of the maps so they can be compared with the faults and volcanic centers. In particular, the Mesozoic rift basin, the Buluk Fault zone and any N-S oriented pre-existing structures that may have been exploited by Ririba rift faults as implied by text.

We fully agree with the Reviewer on this point and –as stated above- we have now introduced a figure in the supplementary material (new Supplementary Figure 5) to show the location and orientation of the pre-existing fabrics characterizing the region.

2. The decoupling between the timing of extension and magmatism is very interesting and important. I think this should be emphasized more in the Broader implications. This is very important for understanding other more mature rift systems. In the absence of other age information, the age of volcanism is sometimes used a proxy for rift activity.

Good point. We have modified the final paragraph (Broader implications) accordingly.

3. The snapshots of dynamical models shown in the main text do not feel very well connected to the observations presented in the paper since they show later stage of extension after the abandonment of the Ririba Rift. I suggest replacing one of the panels in Figure 5 with one of the panels from Suppl Fig 9 that shows the earlier, broader zone of deformation.

OK, we have now modified the manuscript by adding a figure (Fig. 6) to illustrate the setup and results of numerical modelling (including earlier stages of more distributed deformation). We also connected the temporal evolution of the model to the observations in the new Fig. 7. Please also see responses to comments by Reviewer 1 above.

One interesting difference between the numerical models and observations is that during the broad deformation stage, there appear to be many closely spaced smaller faults in the model compared with three relatively localized rift zones in the observations. What explains this difference?

We agree, this is an interesting point. But to be honest it is very difficult to say what causes the difference. We speculate that this could be related to inherited crustal anisotropies that become reactivated in nature. The numerical model can not include anisotropies yet and therefore fails to reproduce this observation. Since this is an extremely speculative point, we decided not to discuss it in the manuscript.

Other minor comments:

Lines 90-92 – Repeated sentence.

Text modified

Line 116 – Give reference for regional plate kinematics

Done

Line 278 – What is meant by “a reduction in plate motion related stresses”?

The sentence has been rearranged and this concept (which was not clear enough) has been removed

Figure 1 – add location of zoomed map to inset in lower right hand corner

Figure modified

-Suppl Fig 1 - It would be nice to plot the displacement profiles for F1 and F2 at the same scale and align them spatially so that its easier to compare patterns of displacement between the two.

Figure modified

Was a similar analysis done on other faults with less ideal orientation for comparison?

Yes, we performed similar analysis on many other smaller (and differently oriented) faults in the area. The results of this analysis are similar to those derived from the analysis of the main border faults of the Ririba rift, as illustrated in the Supplementary Info. These more detailed results will likely be topic of a future work investigating the parameters controlling the characteristics of normal faults in the region.

-Suppl Fig 9 – What are A, B and C labeled in these panels?

This was indicated in the text; anyway, we have now specified this in the modified figure caption.

Reviewers' comments:

Reviewer #1 (Remarks to the Author):

The revised version of Corti and colleagues' paper on the Ririba rift addresses many of my original concerns. In short, the time-frame of faulting and volcanism in the Turkana region is now clearly shown, the seismic gap is explained in their response and their numerical model is setup well and more clearly linked to the new geological results. The paper is much improved and I enjoyed re-reading this version. I would underscore that new results are high-quality and that the scope of the paper is appropriate for Nature Comms.

I have a few issues that ought to be addressed before publication. My only significant concern relates to the age uncertainty on the Hurri Hills volcanics (which constrain the end of the Ririba rifting). While the authors have presented a clearer argument as to why they consider the age of 2.8–2.3 Ma to be the oldest ages of these units, I have two final suggestions/queries

1) These K-Ar ages were generated ~30 years ago and it is not made clear in the main text what the errors on these ages are likely to be. It is well known that there can be large errors in K-Ar ages especially when these are redated by Ar-Ar. While I appreciate that there are difficulties working in this region and that the authors are doing the best they can with the available data I feel they could be clearer on the age uncertainty for these K-Ar dates. At the moment it feels like the ages have been plucked from old literature without much discussion/consideration as to whether they are still robust or not.

The authors need to do their best to evaluate the uncertainty on these ages. They should clarify: 1) whether errors were presented in the original publications (I can't access the 2.8 Ma Charsley age to check); 2) are these errors presented at 2 sigma (so equivalent to their new Ar-Ar ages) and 3) are these ages likely to have changed with updated decay constants (assuming isotopic ratios are presented in these original papers).

2) They show faults at the northern tip of Hurri hills lava pile (shown in Figure 2, supp figure 7) – are these related to Ririba rifting? If so does this not suggest that rifting was continuing throughout emplacement of these volcanics? Its not clear and I could not find any mention of these structures and their implications. A comment is needed.

Minor points

Main text

Please be consistent on spelling of 'Rift' or 'rift' throughout the ms and supplementary files. I have no preference but it must be the same throughout.

L. 98, suggest instead of 'analytical work' you are specific i.e. 'geochronology and geochemistry'

L. 117, could remove 'Overall'

L.124, rephrase, 'The overall fault pattern (excluding the major faults) is characterized by interaction between variably oriented normal faults. This is exemplified at the southern termination ...'

L. 158, are all rocks in the Ririba strictly basalts, it looks like many plot in field of basanite and tephrite in (sup Fig. 8).

L. 169, lavas rather than magmas?

Figure 4. I still don't feel this figures adds much to the main text and could be excluded (as it is shown in Supp Info). If it is included then the ages really need to be given on the figure.

L. 208 – 211, as noted above, error bars really ought to be considered on these ages

L. 220, again what is the variation in the 1.5 Ma timsepan given the errors on the ages?

Figure 5. It is not possible to differentiate between the volcanoes which show Pleistocene activity and Holocene activity (uncertain) as the colors are more or less identical. Suggest change to color scheme. Volcanoes could simply be categorized as Pleistocene activity, Holocene activity and

unknown eruptive history? Could the authors also (approximately) shade in the region where the temporary seismic network of Pointing et al., 1985 picked up seismicity? This would clearly demonstrate that there is no seismic gap. They made a good case in their rebuttal and so it would be good to see this point made clearly in the main text and figures.

Figure 6 d, greyscale strain rate colour bar is not explained in Figure caption or main text. Needs a description and why is the scale different from c?

Figure 7, This is a very helpful figure and good to see the model-data comparison. Can North Lokichar (NL) not be indicated on Figure 1, and mentioned briefly in the introduction. It is a bit confusing to find the first and only mention of this in a figure at the very end of the paper.

L. 367, suggest commas for parenthesis instead of the short dashes

L. 377, 'geologically rapid', '(on timescales < 1Ma)' – keeps age reporting style consistent

Supplementary Information

Section 1, this is a useful addition and really helps to frame the study and provide detail for the keen reader. However, the tone is quite informal in places and it does not feel as well written as the main text. I suggest the authors have another review of this section to improve readability. I won't pick out every instance but a few points I note:

1st line 'a long system of tectonic depressions' very informal length scale needs to be quantified
3rd para 1st line, 'is most likely' rather than 'has been likely', 'variations the' needs to be corrected

4th para, 1st line, 'protracted' instead of 'long-lasting, complex' and again timescales needs to be quantified

4th para also seems to have no references, which is very odd given it is introducing the region

Section 2, 'supports our hypothesis' rather than 'supports our idea'

Supp Table 2, are these rock types correct. Shouldn't MDZ03 be a tephrite not an alkali basalt? Can you also include the totals in the table? This is normal practice and allows a quick appreciation of data quality

Supp Table 3, please use decimal places instead of commas.

Both these tables could use a caption that explains the units, presumably % and ppm, for majors and traces.

Section 6, title 'evidence'

Supplementary Figure 14, as with Fig. 6d could use a description of what the strain rates is showing in lower plot and why the scale is different

Reviewer #2 (Remarks to the Author):

Corti and colleagues present new chronological and geological constraints on the evolution the Turkana depression, and compare their observations with a geodynamical model. The results are convincing and important for understanding rift segment linkage, the role of preexisting structures, and the relationship between tectonism and magmatism in rifts globally and will thus be of interest to a large community. I reviewed an earlier version of this manuscript, and the authors have made a number of changes that address my concerns.

-I previously had difficulty understanding all of the pre-existing structures in this region and how

they influenced different aspects of rift evolution. The authors included a new figure and more text in the supplementary material that better describe these structures and make these relationships clear.

-They have now included more panels on the numerical modeling in the main paper (in particular, panels that illustrate the earlier, distributed rifting phases) which enable a better connection between their observations and the model.

-The new chronograms are also very helpful for visualizing the authors' proposed chronology and comparing it with previous inferences.

I am satisfied with the revisions and have no further comments, and recommend publication.

The detailed responses to the Reviewer's comments are coded as follows:

Reviewer's comments: *italics (11pt)*

Responses: plain text (13 pt)

Reviewer #1 (Remarks to the Author):

The revised version of Corti and colleagues' paper on the Ririba rift addresses many of my original concerns. In short, the time-frame of faulting and volcanism in the Turkana region is now clearly shown, the seismic gap is explained in their response and their numerical model is setup well and more clearly linked to the new geological results. The paper is much improved and I enjoyed re-reading this version. I would underscore that new results are high-quality and that the scope of the paper is appropriate for Nature Comms.

I have a few issues that ought to be addressed before publication. My only significant concern relates to the age uncertainty on the Hurri Hills volcanics (which constrain the end of the Ririba rifting). While the authors have presented a clearer argument as to why they consider the age of 2.8–2.3 Ma to be the oldest ages of these units, I have two final suggestions/queries

1) These K-Ar ages were generated ~30 years ago and it is not made clear in the main text what the errors on these ages are likely to be. It is well known that there can be large errors in K-Ar ages especially when these are redated by Ar-Ar. While I appreciate that there are difficulties working in this region and that the authors are doing the best they can with the available data I feel they could be clearer on the age uncertainty for these K-Ar dates. At the moment it feels like the ages have been plucked from old literature without much discussion/consideration as to whether they are still robust or not.

We agree with the Reviewer on these points and we have improved the manuscript following his/her suggestions. In particular, we have modified both the main text and Supplementary section 6 by adding the errors to the K-Ar datings and discussing them in light of the timing of rift deactivation, as explained below.

The authors need to do their best to evaluate the uncertainty on these ages. They should clarify: 1) whether errors were presented in the original publications (I can't access the 2.8 Ma Charsley age to check);

After several attempts, we have been able to see the original report by Charsley (1987), which indicated the details of the ages shown in the North Horr geological map, including errors (note that a minor portion of these data were also published as supplementary material to the paper by Morley et al., 1992, Journal of the Geological Society, London, 149, 333-348).

All this information is now introduced and discussed in the manuscript (Supplementary info) and illustrated in the modified version of Fig. 2 and Supplementary Figure 7.

2) are these errors presented at 2 sigma (so equivalent to their new Ar-Ar ages)

The report by Charsley and the supplementary material of the Morley paper do not specify if the reported error is 1 or 2 sigma. We looked for the original report on these K-Ar datings by C.C. Rundle (1985, report 85/24) but the British Geological Survey replied us that the report is not available. We were only able to retrieve a similar report by the same Author (C.C. Rundle) for a nearby area (sheet 20; report 85/20) and all the errors therein reported are expressed as 2 sigma; therefore it is very likely that the errors in age of samples from the Charsley report (which were analysed in the frame of the same project – the Samburu-Marsabit mapping project) are similarly expressed as 2sigma. However, since there is no universally established standard as to which confidence level the data should be reported, especially in old publications, it is not possible to be 100% sure of this.

and 3) are these ages likely to have changed with updated decay constants (assuming isotopic ratios are presented in these original papers).

Considering the use of updated decay constants, the change relative to the original set used in Morley et al. (Steiger and Jaeger, 1977) is barely -0.3 % (Renne et al., GCA, 75 (2011) 5097–5100), a shift that is insignificant in the age range explored and, above all, completely over-shadowed by the analytical uncertainties of the K-Ar ages (30-40 % for samples #40 & #41). Also note that our Ar/Ar ages were also calculated using Steiger and Jaeger (Earth Planet. Sci. Lett. 6, 359–362, 1977), so we are actually already comparing on the same basis.

Anyway, it is important to stress here that the uncertainty in the K-Ar dating of the Hurri Hills rocks does not change the results of our work. In particular, although poorly constrained by the K-Ar dating of sample 28 (characterised by a large error of 1.3Ma), the basal flows of the Hurri Hills shield overlie the Bulal basalts, which we have dated (Ar-Ar) at ca. 3.7 Ma, and are overlain by younger products of the Hurri Hills with ages as old as 2.3 Ma (in this case the K-Ar age is better constrained and the error is of 0.17Ma only). Since the basal flows generally seal the Ririba faults (they are only weakly affected by few minor normal faults at the northern termination of the shield) and are overlain by undeformed younger lava flows dated at 2.3 Ma, they mark the end of rift activity (given the uncertainty in the K-Ar dating of the basal flows) to have 'likely occurred close to the Pliocene-Pleistocene boundary', as indicated in the text.

2) They show faults at the northern tip of Hurri hills lava pile (shown in Figure 2, supp figure 7) – are these related to Ririba rifting? If so does this not suggest that rifting was continuing throughout emplacement of these volcanics? Its not clear and I could not find any mention of these structures and their implications. A comment is needed.

The existence of these few, minor faults affecting the basal Hurri Hills lavas was already mentioned in the previous version of the manuscript (Supplementary

Information, Section 6, pag. 21), where we indicated that limited deformation in the area likely occurred during the initial phases of building of this volcanic edifice.

According to the Reviewer's comments, we have now modified section 'Timing of rifting' in the main text and section 6 of the Supplementary information to more clearly illustrate the occurrence of this faulting and to discuss its implications for the timing of rift deactivation in the area. We feel this has contributed to improve the manuscript.

Minor points

Main text

Please be consistent on spelling of 'Rift' or 'rift' throughout the ms and supplementary files. I have no preference but it must be the same throughout.

Many thanks. We prefer to use 'rift', and we have modified it throughout the manuscript and supplementary into.

L. 98, suggest instead of 'analytical work' you are specific i.e. 'geochronology and geochemistry'

Ok, text modified as follows: 'we conducted geological-structural and analytical (geochronological and geochemical) work on samples ...'

L. 117, could remove 'Overall'

Done

L.124, rephrase, 'The overall fault pattern (excluding the major faults) is characterized by interaction between variably oriented normal faults. This is exemplified at the southern termination ...'

Text modified accordingly

L. 158, are all rocks in the Ririba strictly basalts, it looks like many plot in field of basanite and tephrite in (sup Fig. 8).

With the term 'basalt' in previous line 158 we were referring to the basaltic *sensu lato* compositions of the volcanic products of the region, according to previous works.

However, following this comment we indicated in the text (and in Tab. 2 of the Supplementary Material) the compositional classification of our samples according to Le Maitre et al., 2002, "Igneous Rocks. A Classification and Glossary of Terms".

L. 169, lavas rather than magmas?

Text modified

Figure 4. I still don't feel this figures adds much to the main text and could be excluded (as it is shown in Supp Info). If it is included then the ages really need to be given on the figure.

We still prefer to keep this figure, as it summarizes the results of the Ar-Ar dating (whose details are illustrated in supplementary figures 6a-e). Anyway, following the Reviewer's comment, we have now added peak ages (with errors) to the figure.

L. 208 – 211, as noted above, error bars really ought to be considered on these ages

We agree on this and, as explained in detail above, we have now rearranged the sentence accordingly. We have also changed Supplementary section 6.

L. 220, again what is the variation in the 1.5 Ma timespan given the errors on the ages?

As explained above, the errors in ages do not change significantly our results and the 1.5 Ma timespan we indicated. However, we have slightly modified the sentence by changing 'likely' to 'possibly'.

Figure 5. It is not possible to differentiate between the volcanoes which show Pleistocene activity and Holocene activity (uncertain) as the colors are more or less identical. Suggest change to color scheme. Volcanoes could simply be categorized as Pleistocene activity, Holocene activity and unknown eruptive history?

Following this comment, we have now modified the figure. In particular, we now strictly follow the Smithsonian subdivision, and differentiate the volcanoes in Holocene activity and Pleistocene activity only. In order to avoid confusion, we prefer not to introduce the third category of 'uncertain Holocene activity' or similar (which was based on the analysis of the morphology of volcanic vents), also because it should be introduced for a few volcanic edifices only (not central to the region of interest)

Could the authors also (approximately) shade in the region where the temporary seismic network of Pointing et al., 1985 picked up seismicity? This would clearly demonstrate that there is no seismic gap. They made a good case in their rebuttal and so it would be good to see this point made clearly in the main text and figures.

Following the Reviewer's comment, we have now modified Fig 5c to illustrate the region where seismicity has been recorded by the temporary network of Pointing et al.

Figure 6 d, greyscale strain rate colour bar is not explained in Figure caption or main text. Needs a description and why is the scale different from c?

Many thanks for noting this. Both figures 6d and 14 have been now modified by adopting the same strain rate colour scale. We have also modified both figure captions by indicating the use of the semi-transparent strain rate layer.

Figure 7, This is a very helpful figure and good to see the model-data comparison. Can North Lokichar (NL) not be indicated on Figure 1, and mentioned briefly in the introduction. It is a bit confusing to find the first and only mention of this in a figure at the very end of the paper.

Following this comment, we have modified both the Introduction and Figure 1.

L. 367, suggest commas for parenthesis instead of the short dashes

Text modified accordingly

L. 377, 'geologically rapid', '(on timescales < 1Ma)' – keeps age reporting style consistent

Text modified accordingly

Supplementary Information

Section 1, this is a useful addition and really helps to frame the study and provide detail for the keen reader. However, the tone is quite informal in places and it does not feel as well written as the main text. I suggest the authors have another review of this section to improve readability.

Section checked and modified

I won't pick out every instance but a few points I note:

1st line 'a long system of tectonic depressions' very informal length scale needs to be quantified

Text modified

3rd para 1st line, 'is most likely' rather than 'has been likely', 'variations the' needs to be corrected

Text modified

4th para, 1st line, 'protracted' instead of 'long-lasting, complex' and again timescales needs to be quantified

We modified this part and removed this sentence, as it was somehow confusing. We now more clearly introduce the tectonic history of the region, differentiating the Mesozoic-Paleogene and the Oligocene-recent extensional episodes

4th para also seems to have no references, which is very odd given it is introducing the region

References were given in the caption of the figure showing the spatial and temporal relationship of late Tertiary extension and volcanic activity in the Turkana depression (Supplementary Fig. 1). Anyway, according to this comment, these references have been now reported in the first sentence of the 4th paragraph.

Section 2, 'supports our hypothesis' rather than 'supports our idea'

Text modified accordingly

Supp Table 2, are these rock types correct. Shouldn't MDZ03 be a tephrite not an alkali basalt?

Following this comment, we have now classified the rocks according to Le Maitre et al., 2002 ("Igneous Rocks. A Classification and Glossary of Terms") and modified the table accordingly.

Can you also include the totals in the table? This is normal practice and allows a quick appreciation of data quality

Table modified accordingly

Supp Table 3, please use decimal places instead of commas.

Done

Both these tables could use a caption that explains the units, presumably % and ppm, for majors and traces.

Captions added, and units indicated

Section 6, title 'evidence'

Done

Supplementary Figure 14, as with Fig. 6d could use a description of what the strain rates is showing in lower plot and why the scale is different

Figure and caption modified, as explained above.

REVIEWERS' COMMENTS:

Reviewer #1 (Remarks to the Author):

The authors have done a good job answering my final queries and I have no further comments.

They've made the most of the existing geochronological data for the region and have done their best to clarify the uncertainties. This is an interesting paper on a fascinating region and I look forward to reading the final version of the manuscript.

Reviewer comments

No additional comments were raised by the Reviewer.

REVIEWERS' COMMENTS:

Reviewer #1 (Remarks to the Author):

The authors have done a good job answering my final queries and I have no further comments.

They've made the most of the existing geochronological data for the region and have done their best to clarify the uncertainties. This is an interesting paper on a fascinating region and I look forward to reading the final version of the manuscript.

Many thanks